# Middle atmospheric ozone, nitrogen dioxide, and nitrogen trioxide in 2002–2011: SD-WACCM simulations compared to GOMOS observations

Erkki Kyrölä[1], Monika  E. Andersson[1], Pekka T. Verronen[1], Marko Laine[1], Simo Tukiainen[1], and Daniel R.  Marsh[2]

[1]Earth Observation Research, Space and Earth Observation Centre, Finnish Meteorological Institute, P.O. Box 503, 00101 Helsinki, Finland
[2]National Center for Atmospheric Research, Boulder, Colorado, USA

*Correspondence to:* Erkki Kyrölä (erkki.kyrola@fmi.fi)

**Abstract.** Most of our understanding of the atmosphere is based on observations and their comparison with model simulations. In the middle atmosphere studies it is common practice to use an approach, where the model dynamics is at least partly based on temperature and wind fields from an external meteorological model. In this work we test how closely satellite measurements of a few central trace gases agree with this kind of model simulation. We use collocated vertical profiles where each satellite
measurement is compared to the closest model data.

We compare profiles and distributions of $O_3$, $NO_2$, and $NO_3$ from the Global Ozone Monitoring by Occultation of Stars instrument (GOMOS) on ENVISAT with simulations by the Whole Atmosphere Community Climate Model (WACCM). GOMOS measurements are from nighttime. Our comparisons show that in the stratosphere outside the polar regions differences in ozone between WACCM and GOMOS are small, within 0–6%. The correlation of 5-day time series show very high cor-
relation 0.9–0.95. In the tropical region 10°S–10°N below 10 hPa WACCM values are up 20% larger than GOMOS. In the Arctic below 6 hPa WACCM ozone values are up to 20 % larger than GOMOS. In the mesosphere between 0.04–1 hPa the WACCM is at most 20% smaller than GOMOS. Above the ozone minimum at 0.01 hPa (or 80 km) large differences are found between WACCM and GOMOS. The correlation can still be high, but at the second ozone peak the correlation falls strongly and the ozone abundance from WACCM is about 60% smaller than from GOMOS. The total ozone columns ( above 50 hPa)
of GOMOS and WACCM agree within $\pm 2\%$ except in the Arctic where WACCM is 10% larger than GOMOS

Outside the polar areas and in the validity region of GOMOS $NO_2$ measurements (0.3–37 hPa) WACCM and GOMOS $NO_2$ agree within -5–25% and the correlation is high 0.7–0.95 except in the upper stratosphere at the southern latitudes. In the polar areas, where solar particle precipitation and downward transport from the thermosphere enhance $NO_2$ abundance, large differences up to -90% are found between WACCM and GOMOS $NO_2$ and the correlation varies between 0.3–0.9. For $NO_3$,
we find WACCM and GOMOS difference is between -20–5% with very high correlation of 0.7–0.95. We show that $NO_3$ values depend very sensitively on temperature and the dependency can be fitted by exponential of temperature. The ratio of $O_3$ to $NO_3$ from WACCM and GOMOS follow closely to the prediction from the equilibrium chemical theory. Abrupt temperature increases from Sudden Stratospheric Warmings are reflected as sudden enhancements of WACCM and GOMOS $NO_3$ values.

# 1 Introduction

The quality of atmospheric modelling is crucial for making reliable predictions for future climate. The minimum quality requirement for any model is that already measured central atmospheric variables can be simulated within reasonable accuracy. The increasing number of global satellite missions since the discovery of the ozone hole offers a good opportunity to compare models with observed data. Various satellite measurements of trace gases are traditionally compared with validating ground based instruments (see e.g. Hubert et al., 2016), but increasingly they are now also compared with each other (see e.g. Hegglin and Tegtmeier, 2017; Tegtmeier et al., 2013). This activity has led to an increasing understanding of the accuracy of satellite measurements and this is an essential ingredient for a model-measurement comparison.

In this work, we make use of the Whole Atmosphere Community Climate Model (WACCM) from the National Center for Atmospheric Research and compare its results to satellite observations from the Global Ozone Monitoring by Occultation of Stars instrument (GOMOS). We concentrate on an atmospheric region ranging from the stratosphere to lower thermosphere (20–100 km) and on three important minor constituents $O_3$, $NO_2$, and $NO_3$ measured by GOMOS.

Ozone is a central chemical element in the middle atmosphere and essential for stopping short wave UV-light from entering into the biosphere. Ozone has diurnal variability, which in the stratosphere is weak, but at 90–95 km nighttime ozone can be an order of magnitude more abundant that during daytime (see e.g. Kyrölä et al. (2010a); Smith et al. (2013)). Measured satellite ozone profiles are validated using ozone sondes and ozone lidars (see e.g. Hubert et al., 2016). Comparisons to other satellite measurements also help to establish the data quality. Nitrogen dioxide, as a member of the odd nitrogen family, participates in catalytic destruction of ozone especially in the upper stratosphere (Lary, 1997). In polar areas precipitation of charged particles creates vast amount of $NO_x$ which has a long chemical lifetime in the polar darkness. When isolated by a stable vortex, enhanced $NO_x$ can descend into the upper stratosphere, which then leads to natural ozone loss when $NO_x$ becomes illuminated by increasing solar light after the winter season (e.g. Seppälä et al., 2007; Päivärinta et al., 2016). Polar $NO_x$ is also enhanced by polar descent from the thermosphere and exceptionally large increases have measured after so-called Sudden Stratospheric Warming events (SSW) where the vortex structure is disturbed (see for example, Hauchecorne et al., 2007; Randall et al., 2009; Smith et al., 2009; Sofieva et al., 2012; Chandran and Collins, 2014). Nitrogen trioxide is a part of the $O_3$-$NO_2$-NO chemistry, it has a very strong diurnal variation at all altitudes and it is almost absent during daytime (see e.g. Hauchecorne et al. (2005).

WACCM is the atmospheric component of the Community Earth System Model (CESM) (Neale et al., 2013). WACCM is a chemistry - climate model spanning the range of altitude from Earth's surface to the lower thermosphere (approximately 140 km) with 88 vertical levels of variable vertical resolution of 1.1 km in the troposphere to 3.5 km above 65 km (Marsh et al., 2013). Horizontal resolution is 1.9 deg. latitude by 2.5 deg. longitude and the model time step is 30 minutes. In the present analysis version 4 of WACCM was run in specified dynamics mode by constraining dynamical fields to Modern-Era Retrospective Analysis for Research and Applications (MERRA) meteorological re-analyses below 1 hPa. Above the stratopause WACCM dynamics are solved in a free running mode, i.e. temperature and dynamic fields are self-determined (although in practice they are still strongly modulated by MERRA). The version of WACCM used in this work includes

chemistry of the lower, D-region ionosphere with 307 reactions of 20 positive ions and 21 negative ions (see Verronen et al., 2016).

WACCM has been evaluated in many model-measurement intercomparison studies. In Eyring et al. (2010, 2013), WACCM's total ozone values and trends were shown to be in reasonable agreement with satellite observations. Total ozone biases from different latitude ranges were between -5.5–2.3%. Comparisons at specific atmospheric conditions have provided more information on the agreement between WACCM trace gas profiles and observations. In Tweedy et al. (2013), the simulated behaviour of the secondary ozone maximum is compared against SABER measurements during a major sudden warming. The behaviour during SSWs was found to be similar while the nighttime ozone amount is generally underestimated by about a factor of two in WACCM. Comparisons of $NO_x$ during polar winter, when $NO_x$ is influenced by energetic particle precipitation, have been made in many studies (Jackman et al., 2011; Funke et al., 2011; Randall et al., 2015; Andersson et al., 2016; Funke et al., 2017). From these studies it seems that WACCM tends to underestimate mesospheric $NO_x$ by a factor of $\sim$4.

GOMOS (Bertaux et al., 2010) was an instrument on the European Space Agency's ENVISAT satellite which was in operation for just over ten years between 2002 and 2012. The measurement method of GOMOS, stellar occultation, uses light from 180 brightest stars allowing global coverage of measurements with good vertical resolution (2–3 km for ozone, 4 km for $NO_2$ and $NO_3$). The occultation method is self-calibrating because the occulted star's spectrum is also measured without the atmospheric intervention and therefore the primary source data for retrievals (i.e., transmissions) are in principle stable. GOMOS measured 880,000 stellar occultations during the lifetime of ENVISAT. Ozone's relatively large abundance makes it quite an easily observable constituent from satellite instruments using optical measurements. GOMOS measurements can be used to retrieve ozone at altitudes ranging from the troposphere to the lower thermosphere. $NO_2$ and $NO_3$ can be retrieved in the stratosphere.

Our comparisons of GOMOS measurements with WACCM simulations will be based on of individual, co-located profile measurements, whereas in many other model-data studies climatological or other average quantities are used. Our method avoids the problem of uneven (in geolocation and time) sampling that accompanies limb and especially limb occultation measurements and distorts climatologies. In the Coupled Model Intercomparison Project (CMIP) and in the more specialised Chemistry-Climate Model Initiative (CCMI) several atmospheric (or more generally earth system) models including CESM/WACCM have been compared with each other and also with observations (see Tilmes et al., 2016; Morgenstern et al., 2017; Eyring et al., 2010, 2013). Most of the interest in these studies is targeted on future climate projections especially in the troposphere. In this work we are interested to see how well a model simulates the whole middle atmosphere from the upper troposphere up to the lower thermosphere in a limited time range 2002–2011.

Our study is structured as follows. In Sec. 2 we introduce the GOMOS instrument and the measurements we are using in this work. In Sec. 3 the main properties of the WACCM model are introduced. The comparison method is introduced in Sec. 4 and individual comparisons of $O_3$, $NO_2$ and $NO_3$ are presented in Secs. 5–7.

## 2 GOMOS measurements

GOMOS was a stellar occultation instrument on board ENVISAT that was operational from 2002 to 2012 (for GOMOS overviews, see Bertaux et al. (2010); ESA (2001), and https://earth.esa.int/web/guest/missions/esa-operational-eo-missions/ envisat/instruments/gomos).GOMOS measured occultations during both day and night. However, here we use only GOMOS nighttime occultations. Measurements made during daytime suffer from scattered solar light, which leads to low signal/noise ratio of the stellar signal. Daytime data have problems especially below 55 km, and the quality depends very strongly on the properties of the GOMOS target star (Verronen et al., 2007). An alternative approach to retrieve ozone during daytime is to use the scattered solar light observed by GOMOS, this method works well in the stratosphere and lower mesosphere (Tukiainen et al., 2011, 2015). But as mentioned above, we restrict our analysis to nighttime occultation data, partly because for ozone they provide an altitude coverage from stratosphere to lower thermosphere.

GOMOS nighttime profiles of $O_3$, $NO_2$ and $NO_3$ are retrieved from the spectral range 248–690 nm. The integration time of the measurements is 0.5 s, which provides an altitude sampling resolution of 0.2–1.6 km depending on the tangent altitude and the azimuth angle of the measurement. The retrieved ozone profiles have a 2 km vertical resolution below 30 km and a 3 km resolution above 40 km, whereas $NO_2$ and $NO_3$ have a 4 km vertical resolution at all altitudes. Details of the GOMOS retrieval algorithms and data quality are discussed in Kyrölä et al. (2010b) and Tamminen et al. (2010). In this work we use GOMOS data from the ESA processing version 6 in a vertically gridded form (for data access, see Sec. 10). We remove data points that have been measured when ENVISAT was located in the region of the South Atlantic Anomaly. The illumination conditions for the GOMOS measurements are determined by two solar zenith angles controlling solar light at the tangent point and at the satellite location. At the tangent point we require that the zenith angle is greater than $104°$. It has been shown that for zenith angles smaller than $118°$ at the satellite position some stray light can be present, but we have not found any discernible change in our results ignoring this restriction altogether. In the GOMOS gridded ozone data there is an ozone-specific flag that screens stars that do not provide sufficient signal-to-noise ratio for reliable ozone retrieval in the mesosphere-lower thermosphere (faint and cool stars). Profiles considered as outliers either in the stratosphere or in the mesosphere are also flagged. We use only those profiles where all three flags are equal to zero. The total number of GOMOS nighttime measurements is then 238 664. For $NO_2$ and $NO_3$ the ozone flags can be ignored and we get 377 881 measurements. The number of measurements peaked in 2004 and declined thereafter due to the problems connected to the steering mechanism of the instrument. During 2005 no measurements were collected from the period between February to May due to this steering problem. Note that the polar regions are not covered by nighttime measurements during summer months. For other latitudes measurements cover all seasons.

The first comprehensive validation of GOMOS nighttime stratospheric ozone (ESA data version 4) against ground-based and balloon-borne instruments was presented in Meijer et al. (2004). The results showed that GOMOS nighttime ozone agrees within a few percent with the correlative data (sondes and lidars) in the stratosphere outside polar areas. An update of this work was issued by van Gijsel et al. (2010) using the ESA software version 5 and results were similar to Meijer et al. (2004). In this work we are using the ESA software version 6. All three versions (4–6) provide very similar results. The version 6 has been under validation in the ESA projects Valid-2 and Multi-TASTE and the validation reports are available from https://earth.esa.

int/web/sppa/mission-performance/esa-missions/envisat/gomos/cal-val/validation-activities. Recent similar validation results can be found from Hubert et al. (2016) and Sofieva et al. (2016). Results show differences to be within $\pm 3\%$ between 20–45 km. Below 20 km GOMOS show increasing positive bias in the tropics, but in this work we restrict analysis to higher altitudes where such bias is not observed. GOMOS and so-called gold standard of satellite ozone profiles, SAGE II, were compared in Kyrölä et al. (2013) and differences within$\pm 4\%$ in 23–55 km were observed when the SAGE II sunrise and sunset occultations were treated separately. The diurnal variation of ozone in the stratosphere and some sunset-sunrise instrumental factors are contributing to these numbers (see also Sakazaki et al. (2015)). Climatological comparisons of several limb viewing satellite instruments including GOMOS are presented in Tegtmeier et al. (2013).

GOMOS is able to measure ozone up to 100 km when stars with sufficiently high effective temperature are used. For mesospheric heights there are no real validation results, but we can get some insight from comparisons to other satellite measurements. In Verronen et al. (2005) GOMOS and MIPAS ozone were found to agree within $\pm 10\%$ in 25–70 km. Similar results were obtained in Ceccherini et al. (2008). SABER and GOMOS were compared in Smith et al. (2008, 2013), which showed that GOMOS nighttime mesospheric ozone values are about 20% smaller than SABER.

GOMOS measurements can nominally be used to retrieve $NO_2$ at altitudes between 25 and 50 km, while in the polar regions altitudes up to about 70 km can be reached during winter months due to higher $NO_2$ concentrations. There is only one publication where GOMOS $NO_2$ measurements have been compared with in-situ measurements. It is the comparison with balloon-borne instruments (Renard et al., 2008), which indicated an agreement within $\pm 25\%$. In addition, several comparisons against satellite-based observations have been made. Verronen et al. (2009) found that GOMOS $NO_2$ values are 10–25% higher than MIPAS. Comparison with ACE-FTS in Sheese et al. (2016) showed better than 10% agreement between 23–30 km and $\sim 25\%$ between 30–45 km. At higher altitudes larger differences were found, but the necessary correction for diurnal variation made results very uncertain. Nitrogen dioxide has a strong diurnal variation with maximum and minimum amounts seen during early night and early morning, respectively (for diurnal cycle from model simulations, see e.g., Brasseur and Solomon (2005); Kyrölä et al. (2010a). Climatological comparison with HALOE can be found in Hauchecorne et al. (2005).

GOMOS retrieval of $NO_3$ covers the altitude range 25–50 km. During daytime $NO_3$ almost vanishes by photolysis but rises quickly after the sunset from the reactions between $O_3$ and $NO_2$ (for diurnal cycle from model simulations, see e.g. Brasseur and Solomon (2005); Kyrölä et al. (2010a)). There are only few $NO_3$ measurements to which to compare GOMOS measurements. GOMOS $NO_3$ have been compared with two balloon measurements in Renard et al. (2008), but with inconclusive results. In Hakkarainen et al. (2012) GOMOS measurements were compared with SAGE III lunar measurements and the agreement was found to be within $\pm 25\%$.

## 3  SD-WACCM-D simulations

WACCM includes the $O_x$, $NO_x$, $Cl_x$ and $BrO$ families and $CH_4$ with its reaction products. The number of reactions is 217 with 59 species. Heterogeneous reactions with three types of aerosols are also included. The model includes orographic and nonorographic gravity waves (see Garcia et al., 2007). The upper boundary temperature condition is given by the MSIS-model

by Hedin (1991). The same model is used to specify O, $O_2$, H and N upper boundary conditions. At the lower boundary observations are used to specify the surface mixing ratios of CFC-gases, $CH_3$, $N_2O$ and other important gases for stratospheric processes. Historical surface concentrations of greenhouse gases were taken from Meinshausen et al. (2011). The solar irradiance is provided by the model of Lean et al. (2005) which takes into account the spectral and flux variations during the solar

cycle. WACCM includes ionisation rates from Solar Proton Events (SPE) and auroral electrons. More details of the WACCM model can be found from Marsh et al. (2013), Smith et al. (2011), and Garcia et al. (2007).

In this work we use SD-WACCM-D version 4, i.e., the model a) includes chemistry of the lower, D-region ionosphere required for detailed EPP simulations (see Verronen et al., 2016) and b) is run in specified dynamics (SD) mode by constraining dynamical fields below 1 hPa to Modern-Era Retrospective Analysis for Research and Applications (MERRA) meteorological

re-analyses (see Rienecker et al., 2011). SD mode allows for realistic representation of atmospheric dynamics making the simulations directly comparable to satellite observations, while the D-region ion chemistry has been shown to improve the polar mesospheric comparisons for many species, including $NO_x$ (Andersson et al., 2016). In order to provide an ion source for the low-latitude D-region chemistry, ionisation due to galactic cosmic radiation is included in our simulations using the Nowcast of Atmospheric Ionising Radiation for Aviation Safety (NAIRAS) model (for details, see Jackman et al., 2016). For

this study, we also include the ionisation due to 30–1000 keV radiation belt electron precipitation in the energetic particle forcing. For details on the precipitation model and ionisation rate calculation, see van de Kamp et al. (2016). In this energy range, electrons add to $HO_x$ and $NO_x$ production in-situ at 60–90 km altitude, directly affecting mesospheric ozone chemistry at geomagnetic latitudes between $55°$ and $72°$ (Matthes et al., 2017; Andersson et al., 2018). The ionisation rates are applied in WACCM as daily, zonal mean values which depend on the geomagnetic $A_p$ index and latitude.

**4   Comparison method**

In order to compare GOMOS vertical profiles with WACCM simulations each satellite measurement is paired with the closest WACCM latitude-longitude-time profile (i.e., no interpolation between different WACCM grid cells is done). The geolocation of the satellite measurement is defined by the average value when the line-of-sight of the instrument is between the altitudes 20–50 km. In this study, we compare all GOMOS nighttime measurements from 2002 to 2011 to a WACCM simulation run for

the same period. For the satellite measurements the comparison is complete in the sense that every measurement finds its model partner with very good co-location limits: Latitude difference smaller than 0.95 deg., longitude difference smaller than 1.25 deg, and time difference shorter than 15 min. This method avoids the problem of uneven (in geolocation and time) sampling that accompanies limb and especially limb occultation measurements and which may distort trace gas climatologies and their comparisons.

A retrieved GOMOS constituent profile is given at the measurement's refracted line-of-sight altitudes that vary from one measurement to another. In this work we interpolate (linearly) the profiles to a regular geometric altitude grid with one km step. GOMOS constituent abundances are given in number densities. WACCM runs on a pressure grid and abundances are mixing ratios. In order to compare satellite measurements with WACCM we need either to change satellite measurements

to the pressure grid of WACCM or to change WACCM results to the altitude grid used by satellite data. We have selected to work using the WACCM's pressure grid. Therefore, every GOMOS measurement is interpolated to the altitudes obtained from the geopotential heights of the WACCM's latitude-longitude cell nearest to the satellite measurement at the time of the measurement. This brings the number densities of satellites to the pressure grid of the model. In this work we show results in mixing ratios as they more suitable for illustrating results. The transformation to mixing ratios is accomplished by the neutral density distribution of WACCM (originating in the SD-version from MERRA and internal dynamics).

The method we use for comparing collocated satellite and WACCM profiles and their differences at each altitude $z$ is to calculate the bias over a suitable number of profiles in a selected region (time and geolocation) as

$$B(z) = \langle f_{\mathrm{k}}^{\mathrm{W}}(z) - f_{\mathrm{k}}^{\mathrm{G}}(z) \rangle, \tag{1}$$

where $f_{\mathrm{k}}^{\mathrm{W}}$ denotes WACCM and $f_{\mathrm{k}}^{\mathrm{G}}$ GOMOS collocated vertical profiles. Satellite gridded profiles have some missing data from flagged data points or from restrictions of the altitude coverage of measurements. The corresponding WACCM data points are ignored in the average in order to preserve the complete correspondence of the data sets. For practical reasons we will also use the bias in a relative sense as

$$\Delta(z) = 100\% \frac{B(z)}{\langle f_{\mathrm{k}}^{\mathrm{G}}(z) \rangle}. \tag{2}$$

The scaling factor (denominator) is calculated from GOMOS in the same region as the bias.

Calculation of the average estimates is based on dividing spatial and temporal extensions to suitable scales. We average data within 10 degrees in latitude and use zonal averaging. For the polar regions we also show results from a larger latitudinal range (from 60 to 90 degrees south and north). In the time domain the analysis is based on 5-day time averaging in order to capture fast polar processes while keeping reasonable statistical accuracy.

The average from the averaging region and period of time is done by first making averages for each available star (we require at least 10 measurements from each star) and then averaging over the stars involved. This provides more equal contribution from different latitudes covered and no star can dominate the average by its high number of measurements. We apply a median filter ( $|x - \mathrm{median}(x)| > 3 \times 1.4826 \times \mathrm{median}(|x - \mathrm{median}(x)|)$ ) for the distribution of GOMOS values from any given star at each altitude. Any GOMOS outlier means that it and its paired WACCM data are removed. For ozone the number of outliers is less than 1% except at 0.01hPa (ozone minimum) and at the polar latitudes where the number of outliers can reach 5%. For $NO_2$ and $NO_3$ the number of outliers is about 1% and up to 5% in the polar areas. All averages are calculated using the median estimator. After eliminating flagged data and applying minimum number limits we have 231 923 ozone, 358 738 $NO_2$ and 317 653 $NO_3$ WACCM-GOMOS pairs in our comparisons (note that near the upper and lower altitude limits of the GOMOS retrievals the actual number of pairs is usually smaller).

From the WACCM and GOMOS 5-day time series we calculate the WACCM-GOMOS mission average biases and the (Pearson) correlation coefficients $C(z)$. In this step we require that at least 5 time steps are included. This eliminates the latitude belt 80°S–90°S altogether. Notice that the time coverage of the polar latitudes is strongly restricted by the solar zenith condition (nightime) applied on the GOMOS data. In the Antarctic 60°S–90°S the coverage is from mid-February to September and in the Arctic 60°N–90°N from mid-September to mid-April.

In addition to the general data collection rules already explained we have paid special attention to the validity limits in altitude for GOMOS data. This work includes nighttime measurements from 138 stars and each of them have their own valid, constituent specific retrieval range. The GOMOS data we are using include already star specific valid altitude limits for all three gases of this work. These limits are based on yearly averages. In order to handle rapidly changing events we need more

dynamic determination of the validity ranges. Therefore, in this work we have calculated for each star, gas, latitudinal zone and time window (5-days) the average t-value profile (the median value divided by its uncertainty, see e.g., Eq. (1) in Kyrölä et al. (2010a)). We reject those portions from the average profile that have $t < 2$ (this also eliminates negative density averages even if negative individual values are accepted).

An average profile that passes the t-value criterion usually forms a continuous chain of density values (with $t > 2$) in altitude

and the rejected values (with $t < 2$) are located at the low and high altitude parts of the profile. Sometimes two or more disconnected $t > 2$ regions are also present. These regions may represent the real atmospheric state or they can be generated by noise. In the ozone minimum region around 0.01 hPa (80 km) density values are so small that $t > 2$ condition is not usually achieved but t-values recover again at higher altitudes. This minimum structure seems to be omnipresent and we will always include the minimum region in our ozone comparisons. In the polar regions large $NO_2$ values above the normal validity range

of $NO_2$ are observed after a solar storm has hit the Earth. This extension of the profile is short-lived and we apply t-test to monitor its upper limit.

Disconnected noise generated $t > 2$ regions are typically found at altitudes where the density of a retrieved gas approaches to zero. When the density decreases the WACCM's distribution of density values (from an averaging domain) changes from an approximate normal distribution (natural variation) to a nearly lognormal-type of distribution because of the physical lower

limit zero in the model. The GOMOS retrieval approach does not limit the retrieved gas values by a positivity condition as this could lead to bias. As the density approaches to zero the GOMOS' distribution of density values remains nearly normal covering also negative values. Ideally this distribution would settle down around zero with $t \sim 0$ and with the width given by the noise in data. Unfortunately, sometimes this does not happen and we see the distribution average to be positive with $t > 2$. These "ghost" detections may, for example, be generated by the interference of the other gases retrieved at the same time. As a

precautionary measure against these ghosts we reject those altitudes where the GOMOS distribution (from a given star, region, time, altitude) includes more than 20% negative values. For polar latitudes we apply a more relaxed limit of 33%, which allows our analysis to capture fast developing processes.

The procedures explained prevent GOMOS average densities to obtain values too close to zero whereas corresponding WACCM averages are not constrained. For ozone the lowest values are obtained from the ozone minimum and they are about

0.05 ppm for both WACCM and GOMOS. $NO_2$ is removed from the lower Antarctic stratosphere during July-August before the Antarctic ozone hole. The lowest WACCM values (in the present work) are about 0.000015 ppb whereas at the same altitudes the lowest GOMOS values are about 0.04 ppb. For $NO_3$ at low altitudes WACCM shows 0.4 ppt whereas GOMOS 1.7 ppt.

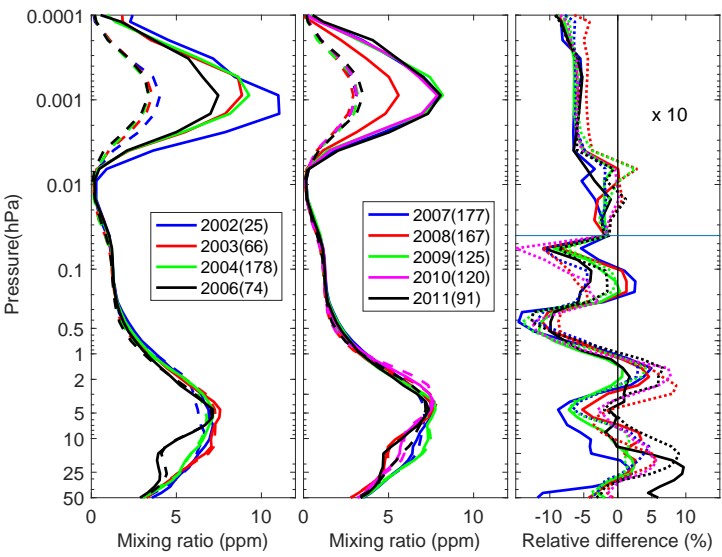

**Figure 1.** Ozone yearly median mixing ratio profiles and median relative differences from GOMOS Sirius occultations and from paired WACCM profiles from 2002 to 2011 in the latitude band 40°S–60°S. Occultations are taking place during late August- mid September. The vertical axis is pressure. Left and middle panels: GOMOS profiles by solid lines and WACCM profiles with dashed lines. The colour coding in the legend boxes shows the measurement year and in the parenthesis the number of measurements. Right panel: Relative median difference WACCM-GOMOS/median(GOMOS). Above 0.04 hPa differences are divided by 10. The colour coding follows left and middle panels, but 2007–2011 lines are dotted.

## 5 Ozone

As an example of retrieved satellite ozone profiles and paired WACCM profiles, we show in Fig. 1 observations from the brightest star in the sky, Sirius. It provides the best signal-to-noise ratio at all wavelengths of GOMOS stellar occultations. These measurements were taking place every year from late August to mid-September. In Fig. 1 we show the yearly median

5 profiles from both the GOMOS observations and the WACCM simulation. It is evident that the observations and the model simulations generally agree well at all altitudes except in the neighbourhood of the second ozone peak (around 0.001 hPa, 91 km) where large differences and yearly variations are evident. The mission average 2002–2011 relative uncertainty of the GOMOS and WACCM Sirius profiles is better than 2% in the altitude range 0.05–50 hPa. The relative uncertainty grows to 10% at and around the ozone minimum at 0.01 hPa, but it reaches again 2% at the second peak and diverges at altitudes above.

10 The WACCM-GOMOS relative difference stays inside ±10% between 0.05–50 hPa, but increases up to 60% at the second peak and grows still at higher altitudes. Differences are statistically sound in the mesosphere whereas in the lower atmosphere the differences fluctuate on both sides of zero.

In order to get a more comprehensive view of WACCM-GOMOS differences for all latitudes we consider now ozone profiles from all eligible GOMOS occulted stars. Profiles flagged by the ozone flags are not included, but all others are included by

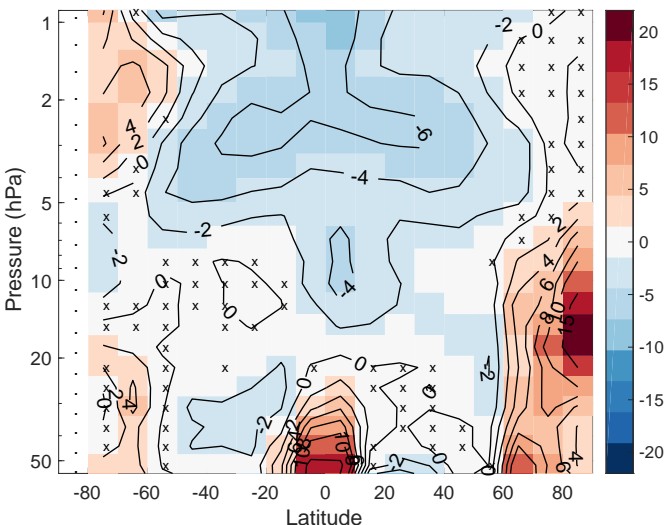

**Figure 2.** The median relative difference (WACCM-GOMOS)/median(GOMOS) of the ozone mixing ratio (in %) in the stratosphere over 2002–2011. Latitudes are from –90° to +90° with 10° resolution. A crossed cell marks a point where the difference does not deviate from zero in a statistically significant way. A cell with a dot marks a point where there are no collocated profiles.

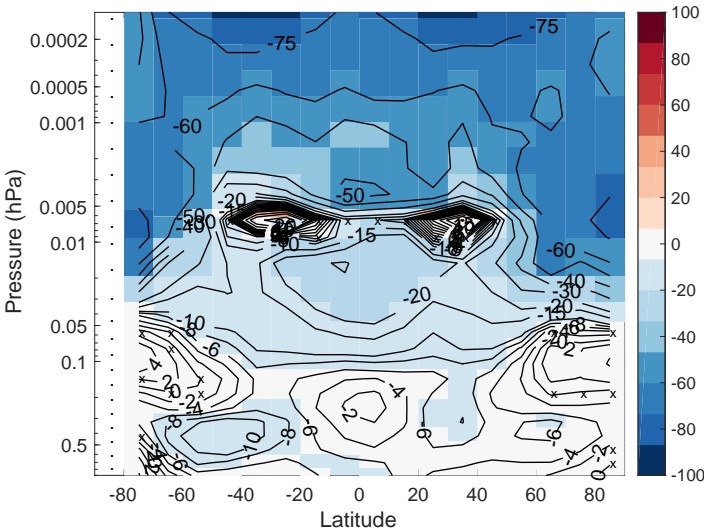

**Figure 3.** The median relative difference (WACCM-GOMOS)/median(GOMOS) of the ozone mixing ratio (in %) in the mesosphere over 2002–2011. Latitudes are from –90° to +90° with 10° resolution. A crossed cell marks a point where the difference does not deviate from zero in a statistically significant way. A cell with a dot marks a point where there are no collocated profiles.

those pressure levels that pass the t-value and the distribution positivity criteria discussed in Sec. 4. Both WACCM and GOMOS main ozone maxima are at the Equator at 10.3 hPa. GOMOS maximum is 9.7 ppm and WACCM 9.4 ppm (difference 3%). In the mesosphere-thermosphere the second mixing value maximum is at the Equator where GOMOS mixing ratio is 10.5 ppm at 0.0005 hPa (94 km) and WACCM 4 ppm at 0.0009 hPa (91 km). The ozone minimum is located at 0.009–0.015 hPa with minimum values above 0.1 ppm. (Notice that WACCM's coarse pressure grid makes altitude estimates uncertain in the mesosphere-thermosphere). The altitude-latitude relative difference distribution between GOMOS and WACCM as a median average of 5-day time series from 2002 to 2011 is shown in Fig. 2 for the stratosphere and in Fig. 3 for the mesosphere-lower thermosphere. The validity range that applies to all latitudes is from 0.00012 hPa to 85 hPa (about 16–105 km). The lower limit in Fig. 2 is taken as 52 hPa (about 20 km) in order to eliminate the GOMOS positive bias below 20 km in the tropics mentioned in Sec. 2. In both figures the differences are mostly statistically significant, points where the WACCM-GOMOS difference is insignificant are marked by crosses.

In the stratosphere outside the polar latitudes WACCM-GOMOS differences are generally small, WACCM values being 0–6 % smaller than GOMOS. This exceeds slightly the $\pm 3$ uncertainty estimates of GOMOS ozone. Larger differences are seen in the tropical lower stratosphere and in the Arctic. In the tropics in the lower stratosphere we see that WACCM values are larger, up to 20%, than GOMOS. In the Arctic between 1–6 hPa WACCM-GOMOS differences are small, between 6–50 hPa WACCM is clearly larger than GOMOS, up to 20% difference at 15 hPa. In the Antarctic the differences are inside -4–+6%.

Figure 3 shows differences in the mesospheric-lower thermosphere, which are moderate up to the altitude 0.05 hPa or even up to the altitude 0.005 hPa outside the polar latitudes. Around 0.1 hPa in the polar areas WACCM and GOMOS agree within $\pm 5\%$. During wintertime a so-called tertiary ozone peak appears in this region (see e.g. Marsh et al., 2001; Degenstein et al., 2005; Sofieva et al., 2009). In the upper mesosphere differences grow strongly and WACCM values are about 60% smaller than GOMOS around the second ozone peak. This result is in agreement between earlier comparisons Tweedy et al. (2013); Smith et al. (2014), where WACCM was compared with MIPAS and SABER measurements. A similar model-measurement difference has been seen in a HAMMONIA model study (see Schmidt et al. (2006)). The GOMOS retrieval is very straightforward in the mesosphere-lower thermosphere and we have not been able to identify any potential sources of uncertainty that could lead to such a large error in the GOMOS retrieval or data. Notice that GOMOS data uncertainty is large at the ozone minimum and the relative difference varies from positive to negative

The ten year mission averaged bias is, of course, a narrow measure on the compatibility of WACCM and GOMOS. We now investigate how WACCM and GOMOS ozone values develop in time. Fig. 4 shows the correlation coefficient of WACCM and GOMOS from 5-day time series as a function of the altitude and latitude. In the stratosphere the correlation is very high, typically 0.85-0.95. At altitudes between the stratopause at 1hPa and the ozone minimum at 0.01 hPa the correlation almost vanishes. High values are seen again between 0.01–0.001 hPa, but the final decrease starts just below the second ozone peak.

Fig. 5 shows the comparison of the WACCM and GOMOS ozone mixing ratio 5-day time series from three latitude bands and at two pressure levels from August 2002 to January 2005. The top panel shows the second maximum where a large bias between the WACCM and GOMOS is evident. Ozone in all three latitude bands shows semi-annual oscillations. WACCM and GOMOS correlation is highest 0.74 at the Equator, 0.54 in 50°S–30°S and 0.35 in 30°N–50°N. At the lowest altitude in the

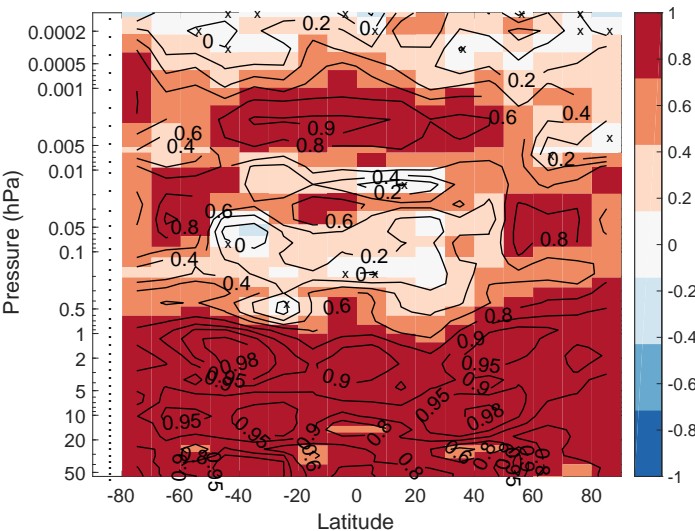

**Figure 4.** WACCM and GOMOS ozone mixing ratio correlation over 2002–2011. The correlation is calculated from 5-day time series. Latitudes are from -90° to 90° with 10° resolution. A crossed cell marks a point where the correlation does not deviate from zero in a statistically significant way. A cell with a dot marks a point where there are no collocated profiles.

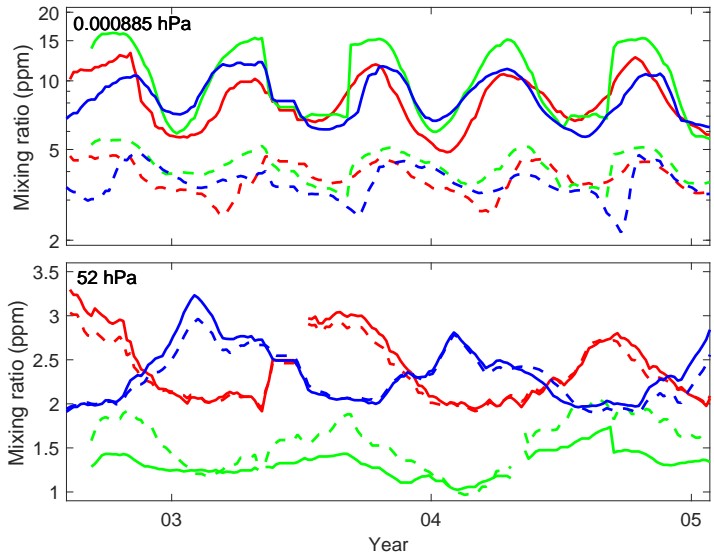

**Figure 5.** WACCM and GOMOS ozone 5-day time series 1.8.2002–31.1.2005. Three latitude belts are shown: 50°S–30°S (red lines), 10°S–10°N (green) and 30°N–50°N (blue). GOMOS values are shown by solid lines, WACCM by dashed lines. The 5-day time series are smoothed by a moving average of 45 days. Note that in the top panel the y-axis is logarithmic.

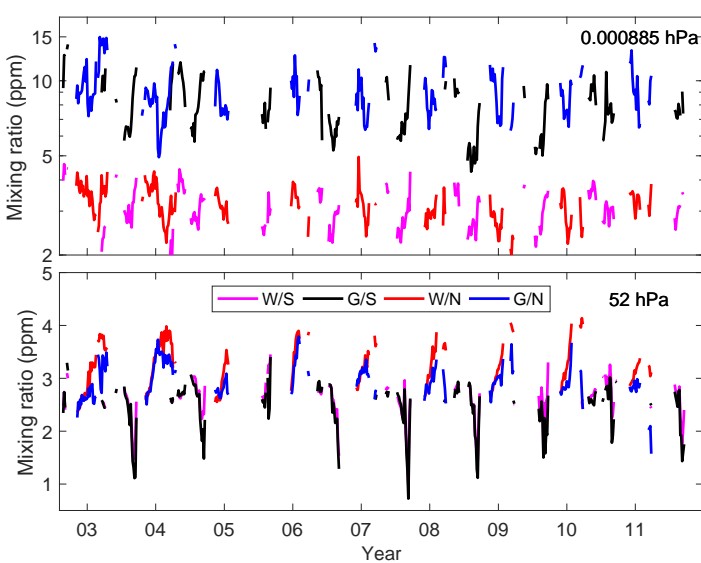

**Figure 6.** WACCM and GOMOS ozone mixing ratio 5-day time series from 2002–2011 in the Arctic 60°N–90°N and in the Antarctic 60°S–90°S. In the top panel the y-axis is logarithmic. The colour coding symbols: W/S, W/N=WACCM in Antarctic, Arctic, G/S, G/N=GOMOS in Antarctic, Arctic.

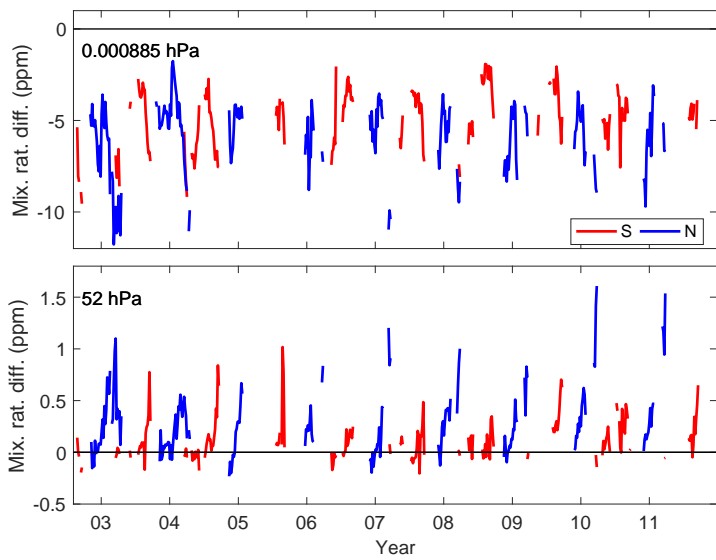

**Figure 7.** WACCM and GOMOS ozone mixing ratio difference from Fig. 6 in the Arctic 60°N–90°N and in the Antarctic 60°S–90°S. The colour coding symbols: S=Antarctic, N=Arctic.

bottom panel we can see that WACCM values in the tropics are consistently higher than GOMOS resulting to the positive tropical bias in Fig. 2 whereas at mid-latitudes there is a good agreement. Correlations are high, 0.83 at the Equator and 0.94 in South and 0.95 in North..

In Fig. 6 we show the 5-day ozone mixing ratio time series in both polar regions at same altitudes than in Fig. 5. The Arctic and Antarctic time series can be shown in the same plot because GOMOS nighttime coverage in these regions is almost complementary in time. Differences are shown in Fig.7. The highest altitude in Fig. 6 (top panel) shows again large differences of the second peak values (in both cases WACCM is on average 62 % smaller than GOMOS). WACCM-GOMOS correlation is 0.59 in the Antarctic and only 0.35 in the Arctic. The bottom panel shows results at the lower end of the valid ozone range. The average WACCM-GOMOS difference is 2.8% in the Antarctic and 8.3 % in the Arctic and correlations 0.89 and 0.62, respectively. In the Antarctic both WACCM and GOMOS show strong ozone reductions, but GOMOS reductions are generally larger. In the Arctic WACCM ozone values are as a rule considerably larger than GOMOS. This tendency continues to higher altitudes and 'explains' the positive peak found in Fig. 2. The exceptionally large ozone loss in 2011 (see Manney et al., 2011) is clearly seen in GOMOS data, but not so clearly by WACCM. A similar even larger difference can be seen in 2010 but now without a real large reduction of ozone.

## 6 Nitrogen dioxide

We start again in Fig. 8 with GOMOS profiles from the Sirius occultations in the latitude band 40°S–60°S. The average uncertainty of the WACCM and GOMOS median profiles is better than 5% in 40–0.5 hPa. The relative WACCM-GOMOS difference is -10–+20% in 40–0.5 hPa. Around the maximum 5 hPa the difference is within $\pm$ 3%. The yearly variation in profiles and differences is large. The reason for this variation is the location of Sirius occultations near the Antarctic vortex where sporadic $NO_2$ enhancements are not totally contained in the polar region.

In Fig. 9 we show the median relative difference between WACCM and GOMOS as a function of latitude and altitude during 2002–2011. The most conspicuous feature of the figure is the variation of the upper valid altitude limit. In the polar regions GOMOS measurements reach up to near 0.05 hPa (about 65 km) whereas elsewhere the highest altitude is about 0.4 hPa (about 55 km). The all latitude lower limit is 37 hPa (about 21 km). The variation of the upper validity limit is the consequence of the data screening using t-values and the positivity condition of the distribution (see Sec. 4). It is important to keep in mind that the high altitude results from the polar regions are solely coming from the few short living $NO_2$ enhancement events whereas $NO_2$ at the lower polar altitudes is measured by GOMOS during the whole winter season. In the polar areas at high altitudes WACCM values are smaller, by 50–90%, than GOMOS. High GOMOS $NO_2$ values are related to extraordinary events that will be discussed later. Outside the polar areas in the stratosphere WACCM-GOMOS difference varies inside -5–+25%. Except the polar regions, the differences are inside the uncertainty estimates of GOMOS $NO_2$. The mission average of the $NO_2$ mixing ratio maximum is at 1.9 hPa by WACCM and at 2.9 hPa by GOMOS. Maximum values are both around 16 ppb and situated at the Equator. The average values in the polar regions are still much higher, in the Arctic 86 ppb and in the Antarctic 40 ppb, but these are only averages over the winter seasons.

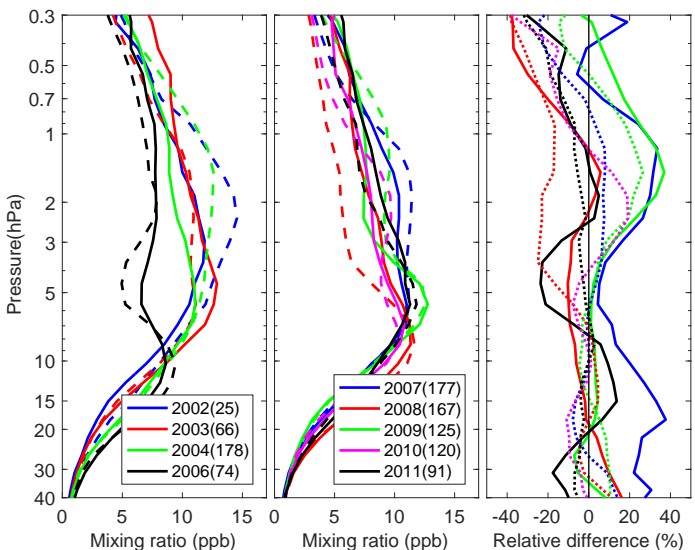

**Figure 8.** NO$_2$ yearly median mixing ratio profiles and median relative differences from GOMOS Sirius occultations and from paired WACCM profiles from 2002 to 2011 in the latitude band 40°S–60°S. Occultations are taking place during late August- mid September. The vertical axis is pressure. Left and middle panels: GOMOS profiles by solid lines and WACCM profiles with dashed lines. The colour coding in the legend boxes shows the measurement year and in the parenthesis the number of measurements. Right panel: Relative median difference WACCM-GOMOS/median(GOMOS). The colour coding follows left and middle panels, but 2007–2011 lines are dotted.

In Fig. 10 we show the WACCM-GOMOS NO$_2$ correlation coefficient's altitude-latitude distribution. In the stratosphere the correlation is high 0.7–0.95 except in the upper stratosphere at the southern latitudes where the correlation vanishes. In the mesosphere at the polar latitudes the correlation varies between 0.3–0.9.

Figure 11 shows WACCM and GOMOS NO$_2$ time series at two pressure levels in the Arctic and Antarctic from 2002–2011.

The differences are shown in Fig. 12. The upper panel in Fig. 11 shows that in both polar regions almost every winter high NO$_2$ events are detected at altitudes much higher than the normal NO$_2$ maximum. Most eminent peaks are taking place during the 2003 Antarctic winter and during the Arctic winter 2003–2004. Elevated NO$_2$ amounts, observed during the winter periods, are known to be generated by particle precipitation events (see e.g. Seppälä et al., 2004, 2007; Funke et al., 2011) and enhanced downward transport of NO$_X$ from the lower thermosphere (e.g. Hauchecorne et al., 2007; Randall et al., 2009; Päivärinta et al.,

2016; Funke et al., 2017). The lower pressure level (the bottom panel) shows the opposite record. The annual oscillation of NO$_2$ has its minimum during the mid-winter. In the Antarctic WACCM NO$_2$ acquires exceptionally low values (in this plot the minimum is 0.0017 ppb) due to denitrification of the lower stratosphere (see e.g., Solomon (1999)). The corresponding GOMOS minimum value is much larger, 0.29 ppb, due to the positivity constraint imposed on GOMOS data.

During the Antarctic winter 2003 a strong increase of NO$_2$ values started in the beginning of June and lasted to mid-

Sepember. This event has been meticulously studied in Funke et al. (2005) using satellite measurements from MIPAS/ENVISAT.

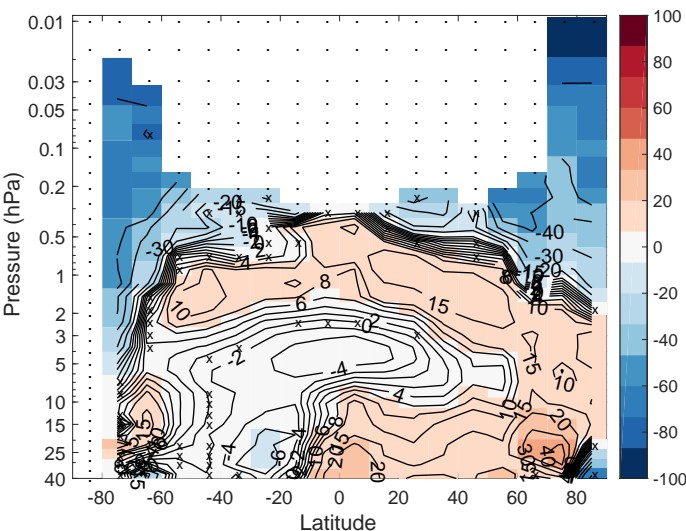

**Figure 9.** The median relative $NO_2$ difference (WACCM-GOMOS)/median(GOMOS)) in % over 2002–2011. Latitudes are from -90° to 90° with 10° resolution. A cross marks a point where the difference does not deviate from zero in a statistically significant way. A cell with a dot marks a point where there are no collocated profiles.

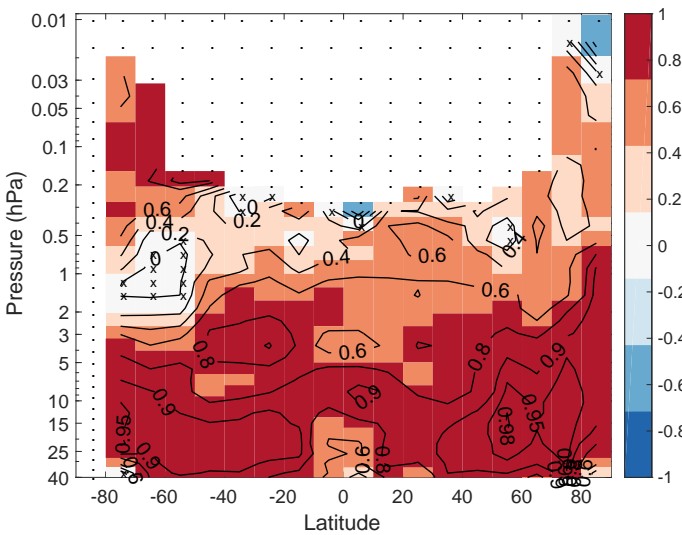

**Figure 10.** WACCM and GOMOS $NO_2$ mixing ratio correlation over 2002–2011. Latitudes are from -90° to 90° with 10° resolution. A crossed cell marks a point where the correlation does not deviate from zero in a statistically significant way. A cell with a dot marks a point where there are no collocated profiles.

The origin of the enhancement is the increase of the $NO_X$ population in the thermosphere by electron precipitation and subsequent downward transport by the meridional transport. In GOMOS data the maximum 5-day median value 134 ppb (at 0.07 hPa) is achieved during 15-19th, July. The corresponding WACCM value is 24 ppb. The Antarctic $NO_2$ enhancement during 2003 is important for two of our earlier results. In Fig. 8 we showed high yearly variation of Sirius $NO_2$ profiles. WACCM

2002 and 2004 profiles around 2 hPa are considerably larger than the corresponding GOMOS profiles whereas during 2003 WACCM and GOMOS profiles agree. This agreement is due to the Antarctic $NO_2$ enhancement during June-September 2003 that peaked before the Sirius measurements were taking place. This extra $NO_2$ lifted GOMOS values to par with WACCM. In Fig. 10 we showed how the WACCM-GOMOS correlation around 1 hPa in the latitudinal range 50°S–80°S is much lower than elsewhere. This correlation (mission average) is dominated by the different temporal development of WACCM and GO-

MOS during June-August 2003 in this latitude region. Around 1 hPa GOMOS values are dominated by the $NO_2$ enhancement whereas WACCM shows the usual annual cycle with the mid-winter minimum. Therefore, a strong anticorrelation emerges between WACCM and GOMOS during the peak of the enhancement event. This anticorrelation is repeated during most of the Antarctic winters, but with smaller amplitude. The correlation over all times sums up for a vanishing correlation. During 2003 the $NO_2$ enhancement and the WACCM-GOMOS anticorrelation extends to non-polar latitudes 50°S–60°S.

Very strong $NO_2$ increases in the Arctic took place between the end of October 2003 and the end of March 2004. This period covers strong proton events on October 28–29, 2003 and November 2–3, 2003 (the so-called Halloween event) and a strong descent period that started in mid-January 2004. The complexity of events is illustrated in Fig. 13 where we show WACCM and GOMOS $NO_2$ mixing ratios and their difference as a function time and pressure. The peculiar ridge form of the distributions is a result from our dynamic GOMOS data selection. Before the Halloween there was not enough $NO_2$ above 1 hPa for GOMOS

to retrieve it. During April this 'normal' level is restored. The elevated $NO_2$ amounts propagate with diminishing peak values down to 3.6 hPa (about 35 km).

It is evident that during the period shown at altitudes above 5 hPa GOMOS $NO_2$ values are most of the time much larger than the ones from WACCM. Figure 13 show how both WACCM and GOMOS capture the enhanced $NO_2$ values around 0.5 hPa, produced by the SPEs in the end of October, and the descent until mid December. WACCM seems to overestimate the

magnitude of this enhancement by 5–20 ppb, which is in agreement with earlier results on $NO_y$ (Funke et al., 2011, Fig. 15). The maximum difference is 39 ppb on 30th October at a pressure level 0.19 hPa. WACCM reproduces only a fraction of the larger increase observed at 0.05 hPa in the beginning of December. This is also true for the strong descent from mesosphere to upper stratosphere observed in January–April. The maximum GOMOS value during these events is 450 ppb at 0.245 hPa as an average over 15th-19th, February, 2004. The corresponding WACCM value is 18 ppb i.e., the difference is 432 ppb.

Mesospheric $NO_2$, and $NO_x$ in general, have been underestimated in WACCM during this period due to a combination of incomplete 1) representation of in-situ production by EEP and 2) recovery from a sudden stratospheric warming in early January, resulting in insufficient descent (see (Randall et al., 2015)).

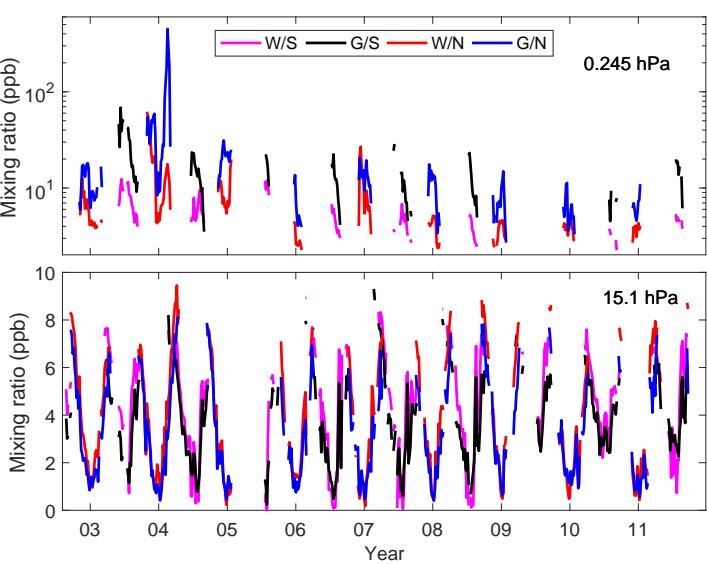

**Figure 11.** NO$_2$ mixing ratio 5-day time series at two pressure levels from the Arctic 60°N–90°N and the Antarctic 60°S–90°S. The colour coding symbols: W/S, W/N=WACCM in Antarctic; Arctic, G/S, G/N=GOMOS in Antarctic, Arctic.

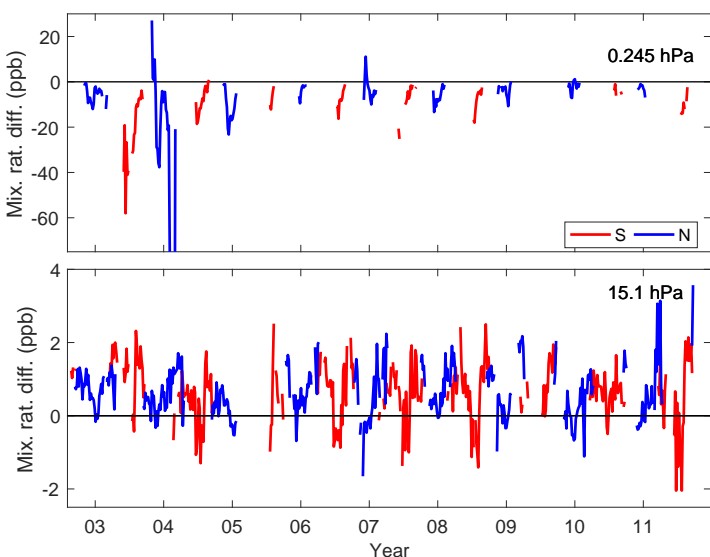

**Figure 12.** WACCM and GOMOS NO$_2$ mixing ratio difference 5-day time series 2002–2011 in the Arctic 60°N–90°N and in the Antarctic 60°S–90°S. The colour coding symbols: S=Antarctic, N=Arctic.

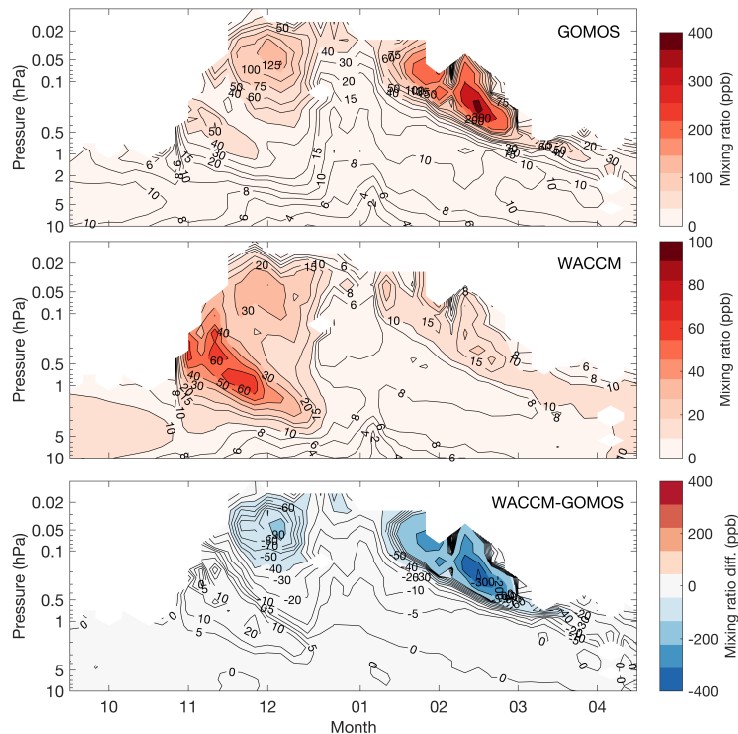

**Figure 13.** NO$_2$ mixing ratio from 5-day time series during 15.9.2003–31.4.2004 from GOMOS (upper panel), from WACCM (middle panel) and WACCM-GOMOS difference (bottom panel) in the Arctic 60°N–90°N. All in ppb-units. Notice the difference in colour scales.

## 7 Nitrogen trioxide

In Fig. 14 we show NO$_3$ profiles from the Sirius occultations in the latitude band 40°S–60°S. The relative uncertainty is better than 10% and the relative difference from -20% to +5% in 1–40 hPa. Near the peak density $\sim$ 2 hPa (40 km) WACCM and GOMOS values are within $\pm$ 2% but at lower altitudes WACCM values are consistently about 20% smaller than GOMOS.

5     The mission averages shows that the general valid altitude region is from 0.7 hPa to 37 hPa (approximately 22–48 km). In the polar regions NO$_3$ values can retrieved up to 0.3 hPa. GOMOS and WACCM NO$_3$ peaks at 2.35 hPa with 270 ppt and in the latitude band 40°S–50°S. The average NO$_3$ values in the polar regions are below 160 ppt. In Fig. 15 we show the median relative differences from 2002 to 2011 between WACCM and GOMOS as a function of latitude and altitude. Around the peak of the NO$_3$ profile the difference between WACCM and GOMOS is typically inside $\pm$5%. This is much better that uncertainty

10   estimates of GOMOS NO$_3$ from validation. In the polar regions, the maximum region excluded, WACCM NO$_3$ is up to 60% smaller than GOMOS.

    In Fig. 16 we show the WACCM-GOMOS NO$_3$ correlation coefficient as a function of the altitude and latitude. Around the NO$_3$ maximum at all latitudes show very high correlations 0.95. The reason for this high correlation is the fact that the

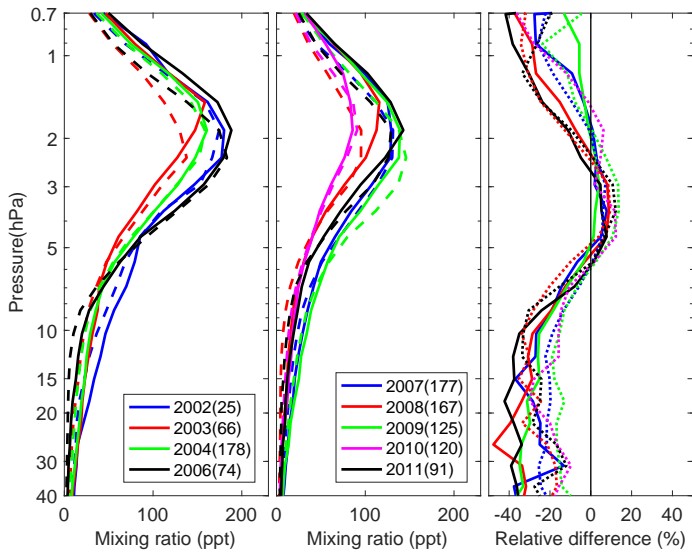

**Figure 14.** $NO_3$ yearly median mixing ratio profiles and median relative differences from GOMOS Sirius occultations and from paired WACCM profiles from 2002 to 2011 in the latitude band $40°S$–$60°S$. Occultations are taking place during late August- mid September. The vertical axis is pressure. Left and middle panels: GOMOS profiles by solid lines and WACCM profiles with dashed lines. The colour coding in the legend boxes shows the measurement year and in the parenthesis the number of measurements. Right panel: Relative median difference WACCM-GOMOS/median(GOMOS). The colour coding follows left and middle panels, but 2007–2011 lines are dotted.

mixing ratio of $NO_3$ is very sensitive to temperature (see Hauchecorne et al., 2005; Marchand et al., 2007; Kyrölä et al., 2010a; Hakkarainen, 2013). When we calculate the correlation of WACCM's $NO_3$ with the model temperature (in the stratosphere MERRA), we get values from 0.7 to 0.99 in the altitude range 2–50 hPa. Similar positive correlation values are seen between GOMOS $NO_3$ and MERRA temperature between 2–5 hPa. Temperature-related issues are a probable cause for the observed
$NO_3$ differences in the polar regions evident in Fig. 15. It is plausible that in the polar regions MERRA underestimates real temperatures except in the neighbourhood of the $NO_3$ maximum. The temporal cycle is correct but the absolute values differ.

Dramatic examples about the temperature dependence of $NO_3$ can be seen in the polar time series of Fig. 17 at 3.7 hPa (this altitude seems to be most sensitive to temperature). In the Arctic, the strongest peaks in mixing ratio are caused by the large changes in temperature during Sudden Stratospheric Warming events (e.g. Sofieva et al., 2012; Butler et al., 2017). In
the Antarctic the $NO_3$ cycle follows the normal annual cycle of the temperature with one exception: During the 5-day period around 28 July 2010 $NO_3$ values have a major jump (for analysis of this case, see de Laat and van Weele (2011)). Note that the famous 2002 SSW in the Antarctica was not captured by GOMOS measurements. It seems that at the sudden warmings (with the Antarctic case excluded) WACCM values considerably exceed the corresponding GOMOS values and we can speculate that MERRA overestimates the real temperature. A detailed evolution of the strong Arctic event in December 2003–January
2004 is shown in Fig.18. WACCM and GOMOS values show similar temporal development, but the actual values differ.

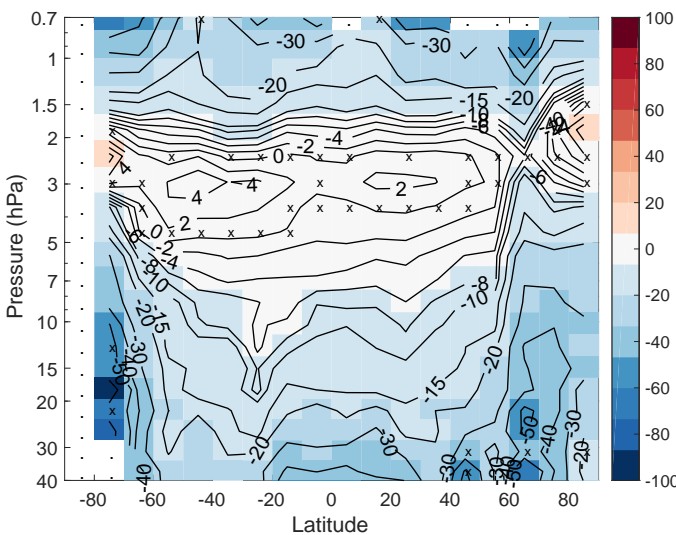

**Figure 15.** The relative $NO_3$ difference (WACCM-GOMOS)/median(GOMOS) in % during 2002–2011. Latitudes are from -90° to 90° with 10° resolution. A cross marks a point where the difference does not deviate from zero in a statistically significant way. A cell with a dot marks a point where there are no collocated profiles.

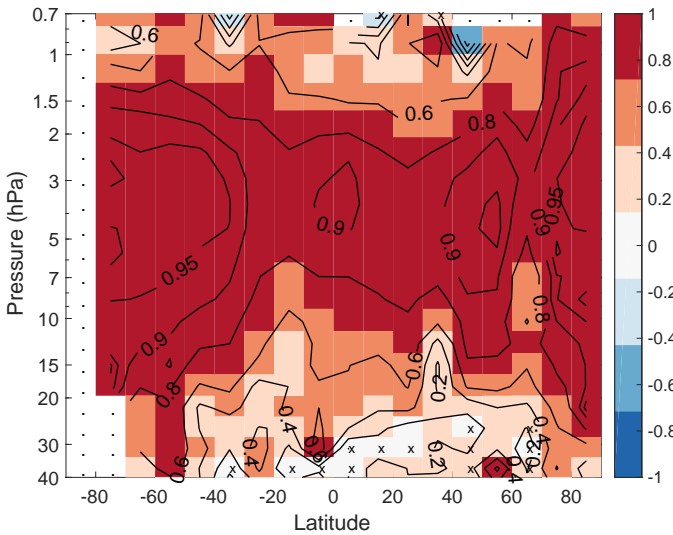

**Figure 16.** WACCM and GOMOS $NO_3$ mixing ratio correlation over 2002–2011. Latitudes are from -90° to 90° with 10° resolution. A crossed cell marks a point where the correlation does not deviate from zero in a statistically significant way. A cell with a dot marks a point where there are no collocated profiles.

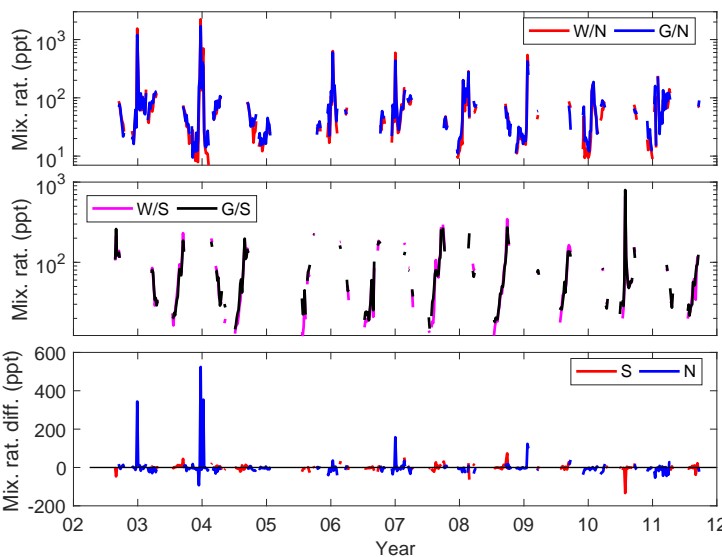

**Figure 17.** $NO_3$ mixing ratio 5-day time series at 3.7 hPa from WACCM and GOMOS from 2002 to 2011 in the Arctic 60°N–90°N (upper panel) and in the Antarctic 60°S–90°S (middle panel). The colour coding symbols: W/S, W/N=WACCM in Antarctic, Arctic, G/S, G/N=GOMOS in Antarctic, Arctic. In the both panels the y-axis is logarithmic. In the bottom panel the mixing ratio difference is shown for the Arctic and the Antarctic in the mixing ratio unit. The colour coding symbols: S=Antarctic, N=Arctic.

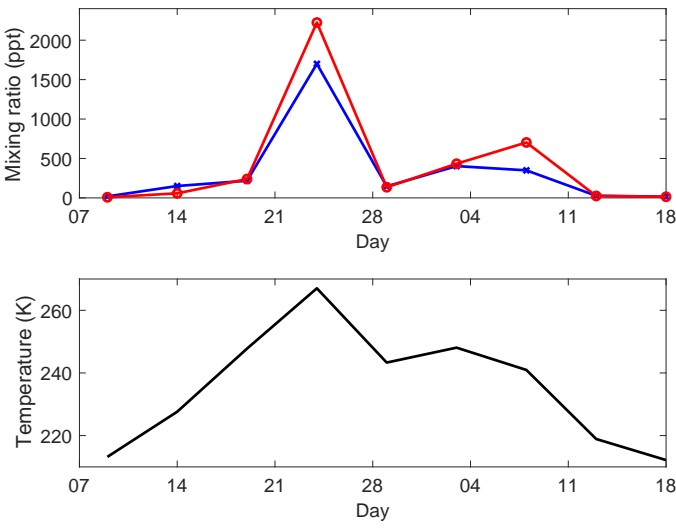

**Figure 18.** The upper panel: WACCM (red) and GOMOS (blue) $NO_3$ 5-day time series 7.12.2003–18.1.2004 in the Arctic 60°N–90°N at 3.7 hPa. Lower panel: MERRA temperature for the same period and altitude.

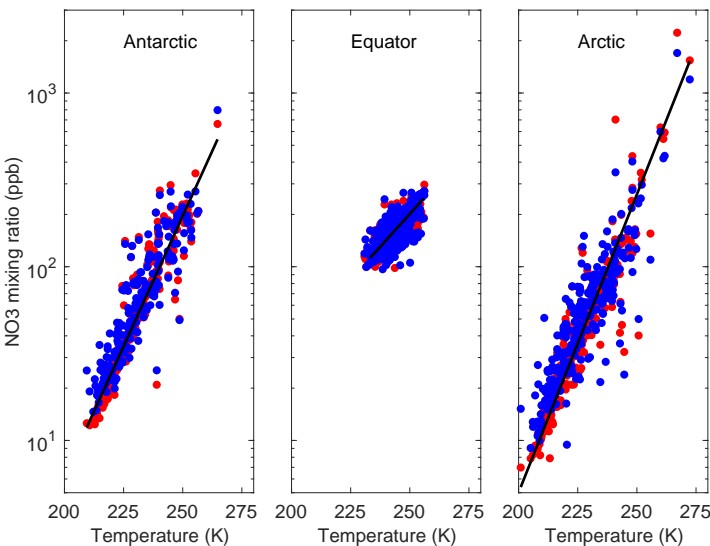

**Figure 19.** $NO_3$-temperature scatter-plot at 3.7 hPa. The left panel: the Antarctic $60°S$ –$90°S$. The middle panel: the Equator $10°S$ –$10°N$. The right panel: the Arctic $60°N$–$90°N$. Red dots are from WACCM and blue dots from GOMOS. Exponential fits are done to temperature gridded WACCM data. Data for all latitudes are from 5-day time series from 2002–2011.

We can further study the temperature dependence of $NO_3$. In Fig. 19 we have plotted WACCM and GOMOS mixing ratio values as a function of MERRA temperature at 3.7 hPa. The dependence on temperature is nearly exponential from both sources in the polar regions. The coefficients of the exponential are 0.069/K for the Antarctic and 0.079/K for the Arctic. The fitting of the equatorial values is more prone to errors as the temperature variation is more limited than in the polar regions.

The two polar coefficients decrease below and above the selected altitude level 3.7 hPa.

In Brasseur and Solomon (2005); Marchand et al. (2004) a formula for the ratio of $NO_3$ to $O_3$ densities is derived assuming nighttime chemical equilibrium. In Fig. 20 we show how this theoretical ratio and the ratio calculated from the WACCM simulated data compare with the ratio determined from GOMOS data. The theory values are calculated using temperature form WACCM. WACCM, GOMOS and the theoretical values show good agreement inside the maximum region of the $NO_3$

mixing ratio excluding polar latitudes. Theoretical values start increasing strongly compared to GOMOS above 1.5 hPa whereas WACCM slightly decrease in the same region. Both WACCM and theory show smaller values with respect to GOMOS below 10 hPa.

## 8 Conclusions

In this work we have compared the state-of-the-art chemistry - climate model WACCM to measurements from the satellite

instrument GOMOS. Measurements cover years from 2002 to 2011 and they are from nighttime. We have compared $O_3$, $NO_2$ and $NO_3$ mixing ratios using 5-day time series. We have also calculated the correlation of GOMOS and WACCM time series.

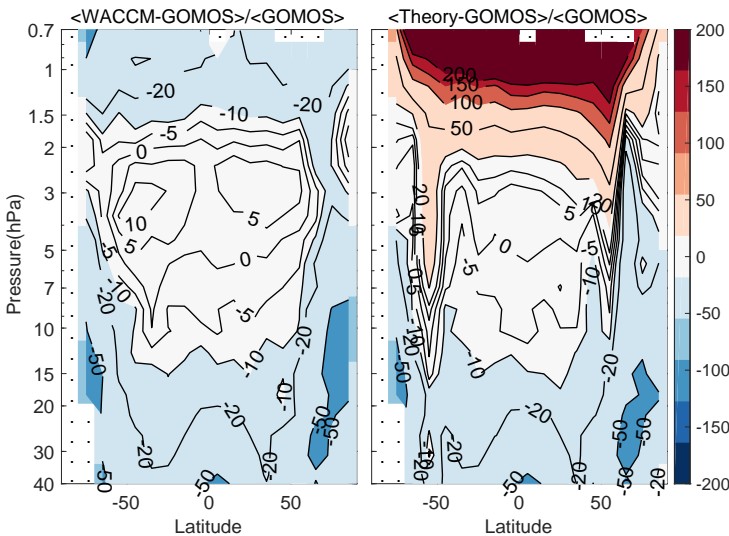

**Figure 20.** The $NO_3/O_3$ ratio from WACCM and from the equilibrium chemistry theory (see Brasseur and Solomon (2005) ) compared to the corresponding ratio from GOMOS. Relative differences. Data are from 5-day time series from 2002–2011. A cell with a dot marks a point where there are no collocated profiles.

The comparisons are done with collocated profiles, which eliminates differences from the natural variability and sampling patterns.

This comparison has required a considerable effort to ensure the quality of the observational data. GOMOS nighttime observations collect photons from 138 different stars varying widely with their luminosity and effective temperature. This
variation causes large differences in the quality of trace gas profiles. For ozone we have used three GOMOS ozone data flags to remove low-quality profiles, for $NO_2$ and $NO_3$ there are no such quality flags available. In order to form reliable average profiles from individual GOMOS trace gas profiles it was necessary to determine upon the altitude limits of valid data in profiles. In the present work we have determined the limits for all time steps, all latitude bands and for all stars using two criteria. First, we have demanded that for valid altitudes the t-value (average density/uncertainty) is larger than 2 and second,
that the distribution of GOMOS values is located mainly on positive density values. This approach has produced altitude limits of valid data that earlier have been estimated using a priori knowledge.

Our comparisons show that in the stratosphere (1–50 hPa) outside the polar regions WACCM ozone values are are 0–6 % smaller than GOMOS values, which slightly exceeds the uncertainty estimates of GOMOS measurements. The difference patterns are consistent in time during 2002–2011. In the tropical region in the lower stratosphere WACCM measurements show
consistently larger values (up to 20%) than GOMOS. In the Arctic GOMOS measurements show smaller ozone values (up to 20%) than WACCM. In the Antarctic the ozone hole evolution is in better agreement. In the mesosphere above the ozone minimum at 0.01 hPa (or 80 km) large differences are found between WACCM and GOMOS. Differences exist in the values

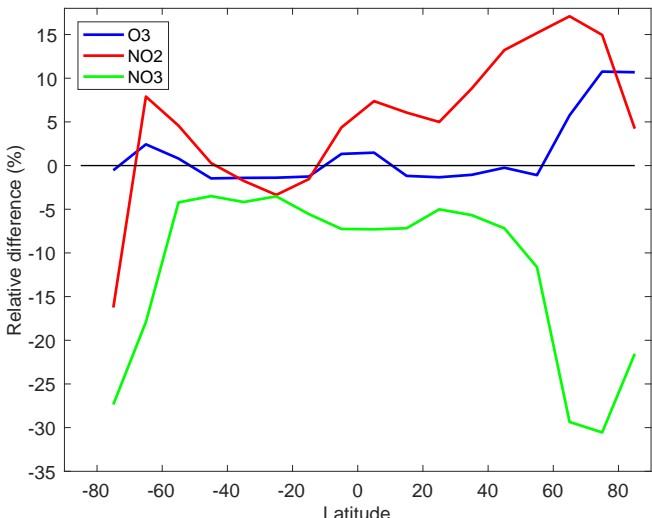

**Figure 21.** The relative difference of WACCM and GOMOS vertical columns of ozone, $NO_2$ and $NO_3$. The vertical extent of the column is 0.0002–50 hPa for ozone, 0.4–37 hPa for $NO_2$ and 1.1–26 hPa for $NO_3$.

of the mixing ratio and also in the correlation of time series at the second ozone maximum. Differences may be connected to WACCM's temperatures in the mesosphere or to specific parameter values that control the gravity wave dissipation in WACCM (see Smith et al. (2014)). The correlation of GOMOS and WACCM time series is high except in the non-polar region in the mesosphere just below the ozone minimum and at the altitudes from the second ozone maximum and above.

Outside the polar areas and in the validity region 0.4–37 hPa WACCM and GOMOS $NO_2$ values agree reasonably well. In the polar areas, where solar particle precipitation and downward transport from the thermosphere enhance $NO_2$ abundances, GOMOS values are much larger than WACCM. The correlation of time series is moderate in the stratosphere except in the upper stratosphere at the southern latitudes where $NO_2$ downdraft events cause anticorrelation between WACCM and GOMOS . GOMOS measurements and simulation by the new version of WACCM are in better agreement for the direct particle initi-

ated $NO_2$ increases, but for the downdraft cases GOMOS values are much larger than the ones from WACCM. The overall correlation of the polar 5-day time series is still quite high in the middle atmosphere.

For $NO_3$, we find WACCM values agree largely with GOMOS. In the validity region 1.2–5 hPa the correlation is very high. Because the $NO_3$ abundance is controlled by temperature, the WACCM-GOMOS $NO_3$ difference can be used as an indicator about the accuracy of MERRA temperature information. We found that $NO_3$ temperature dependence can be fitted reasonably

well by an exponential function in the polar regions. The $NO_3$/chem(O-3) ratio follows quite accurately the result from an equilibrium chemical theory.

The differences in trace gas profiles can also be studied by comparing vertical column densities. The vertical columns can be calculated from number densities at geometric heights of the pressure levels. In Fig. 21 we show the relative difference of

WACCM and GOMOS columns. The vertical extent of the column is 0.0002–50 hPa for ozone, 0.4–37 hPa for $NO_2$ and 1.1–26 hPa for $NO_3$. These limits avoid all missing data cases and include the number density maxima of the gases. The vertical ozone column is 208 Dobson units at the Equator (the full vertical column is about 300 Dobson units) and about 145 Dobson units at the poles. The total column for $NO_2$ varies between 0.05–0.17 Dobson units and between 0.0003-0.001 Dobson units for $NO_3$. We can see that GOMOS and WACCM total ozone columns agree within $\pm 2\%$ except in the Arctic where the WACCM column is 10% larger than GOMOS. WACCM $NO_2$ column is uo to 15% larger than GOMOS except at the southernmost latitudes where enhanced $NO_2$ events have deeper penetration than in north. WACCM $NO_3$ columns are -5% smaller outside the polar areas whereas in the polar areas the difference is around 30%.

In this work we have tried to expose agreements and differences between the WACCM model and the GOMOS measurements. To understand underlaying reasons for differences a detailed and presumably difficult analysis of the model physics and chemistry is necessary. Perhaps the only exception is temperature from the external meteorological model that we think is the reason for $NO_3$ differences in the polar regions. On the GOMOS data side, there is still room for better algorithms and more extensive validation especially in the polar regions. A wider comparison including additional relevant constituents from other satellite instruments would help to vindicate our results and to help finding the underlaying reasons for differences.

## 9  Code availability

The SD-WACCM-D model will be available from NCAR. All the WACCM and satellite data have been processed using Matlab-software. The specific routines used in this work can be requested from the first author.

## 10  Data availability

All data can be requested form the first author. Data will be placed on publicly accessible server in due time. The size of the GOMOS-paired WACCM data set is 2.2 Gb. The GOMOS data used in this work is a Matlab version of the so-called user friendly (UFP) GOMOS data. These UFP data (in netCDF-4 format) are available form the ESA data portal https://earth.esa.int/web/guest/data-access/browse-data-products. The collocated Matlab-data sets include WACCM-data and the paired satellite data. The size: 4.8 Gb.

*Competing interests.*  No competing interests.

*Acknowledgements.*  The authors want to thank anonymous reviewers for useful comments and corrections. The work of E.K. was partly supported by ESA's ALGOM-project. The work of M.E.A. and P.T.V. was supported by the Academy of Finland through the project #276926 (SECTIC: Sun-Earth Connection Through Ion Chemistry). D.R.M. was supported in part by NASA grant NNX12AD04G. The National

Center for Atmospheric Research is operated by the University Corporation for Atmospheric Research under sponsorship of the National Science Foundation.

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
