# Peer review of "Middle atmospheric ozone, nitrogen dioxide, and nitrogen trioxide in 2002–2011: SD-WACCM simulations compared to GOMOS observations"

_Atmospheric Chemistry and Physics, 2017_

## Referee Comment (RC1) · Anonymous Referee #1 · 26 Jan 2018

This manuscript presents a comparison between GOMOS observations and WACCM simulations with nudged stratospheric dynamics for ozone NO2 and NO3 profiles in the middle atmosphere. This kind of comparison is very valuable because model outputs are widely used. Models provide information on a global fixed geographic-altitude grid and at any local time, which is not the case for satellite observations, making the use of the data much easier. WACCM is one of the most used models in the atmospheric community. GOMOS is probably the best instrument for mesospheric ozone and the only instrument measuring NO3 in the stratosphere. It is also one of the few instruments acquiring data in the polar night, which allows for instance to follow the descent of mesospheric NO2 layer in the polar winter mesosphere. However GOMOS data are

not easy to use due to the large variety of spectra from the 180 different stars and the irregular geographic coverage of the observations. The methodology applied to select the valid GOMOS data is particularly rigorous, giving confidence to the results. I recommend the publication of the manuscript in Atmospheric Chemistry and Physics after minor revisions as listed below.

- In equation (2), page 7, the scaling factor is computed using GOMOS data $f_{k}G(z)$. It may cause some problems when GOMOS values are small compared to their uncertainty with the problem of negative values within the error bars. Why not to use WACCM data $f_{k}W(z)$ instead?

- Page 7, line 13, from where is coming the factor 3x1.4826 for the elimination of outliers. Does it correspond to 3 sigma in the median statistic? This is not the same factor that the one given in equation (1) in Kyrölä et al. (2010a).

- Page 7, lines 22-27, the discussion on regions defined with $t\_value > 2$ is not clear. Please try to reformulate it.

- Page 13, figure 8, do you have any explanation for the larger interannual NO2 variability in WACCM than in GOMOS? In general observations exhibit a larger variability than model outputs.

- Page 20, line 14, there is an extended discussion in Marchand et al. (2007) on the relation between GOMOS NO3 concentration and temperature with the same conclusion that NO3 is a good proxy for upper stratospheric temperature. Please cite this paper: Reference: Marchand et al., 2007, Temperature retrieval from stratospheric O3 and NO3 GOMOS data, Geophys. Res. Lett., 34, L24809.

- Page 21, figure 20, it would be better to plot the NO3-temperature diagram with NO3 in log-scale in order to show the exponential relation .

---

## Referee Comment (RC2) · Anonymous Referee #2 · 30 Jan 2018

This is a very interesting paper that highlights how comparing satellite measurements and model output can start to help us better understand the middle atmosphere. It is generally well written, however there are a few sections that need some revision for better clarity (listed below). One general concern is that the descriptions of the results are often vague and not quantified. This needs to be addressed before publication, and the list below also includes instances where this is the case. After all the relatively minor concerns listed below are appropriately addressed, I would recommend the paper for publication.

Figures 1, 8, and 15 really should have a third panel that shows all the relative differ-

[Figure]

ence profiles (especially 8 and 15).

NOTE: line numbers aren't accurate in the pdf version that I have, so hopefully that's the case with the authors' version as well and this will all make sense!

P1 Throughout the abstract, many value judgements are made (e.g. reasonable agreement, high correlation, etc.) without quantifying the values. Please backup these claims with the exact values that you have measured.

Line 15 – unclear what is meant by "the validity region"

P2 Lines 5-6 – please briefly explain how these references are examples of improving understanding of accuracy.

Line 14 – Smith et al. 2013 (doi:10.1002/jgrd.50445) would be an excellent ref to include as well.

Lines 14-16 – Mention of validation and comparison studies should include references.

Line 22 – "see" on its own tends to imply a full list, something like "e.g." or "see for example" would be more appropriate

Line 24 – should mention this is at all altitudes.

Line 26 – change to "to the lower thermosphere"

Line 32 – Does this mean that ion chemistry is included? Or just (non ion) chemistry within the D region? Please specify in the text.

P3 Lines 1-5 – tends to be vague. Terms "reasonable agreement," "compares well," and "found to be similar" need specific quantified values in order to back up these judgements. Same with "good" on line 11.

Line 11 – What is meant by brightest?

Line 16 – "mesosphere and" can be deleted

Line 17 – "comparison" should be "comparisons"

Line 27 – "Sec. the" should be "Sec. 3 the"

P4 Lines 1-2 – should be "this approach has"

Line 6 – "in detail" is not needed

Line 7 – "those" is not necessary

Line 12 – should be something more like "there is an ozone-specific flag that screens out stars. . ."

Line 14 – should be "outliers" and "stratosphere"

Lines 14-15 – I assume you're not setting the flags to zero, rather you're only using profiles where flag values are zero.

Line 27 – do you mean "within ±3%" ?

Line 30 – do you mean "within ±4%" ?

P5 5th line (labelled l38) – could reiterate that this is nighttime profiles being compared.

Line 9 – Sheese et al. 2016 (doi:10.5194/amt-9-5781-2016) is the more recent ACE-FTS NOy validation reference and should be added. Seems to show GOMOS ∼0-10% higher between 23-30 km, ∼25% higher at 30-45 km (although seems ACE-FTS has low bias of ∼10% in this region).

Line 25 – what kind of observations?

P6 Line 16 – delete "a"

Line 19 – would be good to add here how many collocated profiles there are

P7 Eqn 2 – could change to "100%"

Line 6 – "WACCM" should be "GOMOS"

Line 10 – "...processes while keeping reasonable..."

Line 11 – is it the Pearson correlation coefficient?

Line 12 – What is meant by "averages over number of"?

Lines 13-14 – The equation doesn't make it clear exactly how the data is being filtered—please rephrase for clarity. Is this method done using all data at a given altitude? Is it done in latitude bins? Also, in atmospheric datasets where the data is very often neither Gaussian nor uni-modal, using the MAD as a filter can often lead to filtering out a lot of inlying data with the outlying data. If you haven't already, please check that this method isn't "over-filtering" your data, and if everything is okay it would be good to specify that this check was done and how much data is filtered out using this method.

P8 Figure 1 – It would be nice if both figures had the same y-axis labels

Caption – Please specify in caption that these are Aug-Sept profiles

Discussion of Figure 1 – It would be worth noting that both GOMOS and WACCM are exhibiting the tertiary peak and are in good agreement in both height (∼68 km, which actually seems a bit low for the tertiary peak, see Degenstein et al 2005, doi:10.1016/j.jastp.2005.06.019) and concentration.

Line 13 – delete "the"

P9 Line 1 – "reaches again 2%" must be a typo. At the secondary peak, the differences clearly much larger.

Fig 2 caption – might want to say "A cell with a dot marks where there are no collocated profiles." Same with Figures 3, 4, 9, 10, 16, and 17.

P10 Line 18 – "in the two cases"

Line 30 – it may be more prudent to something more along the lines of "we have not

been able to identify any potential sources of uncertainty that could lead to such a large error in the GOMOS data"

P11 Figure 5 – At the secondary maximum the WACCM seasonal variation is very difficult to discern and it's not immediately clear that WACCM and GOMOS are in phase. Could this panel have the y-axis on a log scale?

P12 Figures 6 – A legend (maybe on the rhs) would make the plots much easier to read. Same with Figures 7, 11, 12, 18.

Figure 7 – The red and blue is slightly confusing, because the reader will naturally be comparing to Figure 6 where the same colours indicate only Arctic values. Please use two styles of lines that relate better to Figure 6. Same with Figure 12.

Line 34 – "correlation is typically very high,"

Lines 34-35 – I find this sentence somewhat confusing, please rephrase for better clarity.

P13 First paragraph – please include quantification of the differences and correlation for all three panels of Figure 5.

Line 4 (6th line) – "whereas"

Lines 5-12 – Please include quantification of the differences and correlation for all three panels of Figure 6 and 7.

Figure 8 – should include altitude scale to match Fig 1. Same with Figure 15.

P14 First paragraph – are these also Aug-Sept? If so, please mention here and in Fig 8 caption. Same with NO3 results/figure.

Fourth line – "maximum at 5 hPa the difference is within"

Line 9 – "will be discussed"

Line 11 – "is typically 0-10%" and "typically agree within $\pm$5%" as differences do reach

higher values in the respective regions

P15 First paragraph – Please quantify discussion of correlation coefficients

Line 13 – Should start sentence with something like, "As seen in Fig. 13 and 14,. . ."

P16 First line – it's somewhat confusing that you're discussing the difference in ppb when figure 13 is in %, please make this consistent (or discuss both % and ppbv). Unless, you're referring to Fig 14 here, but Fig 14 doesn't show results for 0.5 hPa. Either way, this section needs to be made clearer (as to what Figures you're discussing and what they show).

Line 5 – "peak density, $\sim$2hPa"

Line 10 – "is typically inside"

P17 Figure 13 – units are missing on both panels

Last line – "The secret behind" could be something more like "the reason for"

P18 Fifth line – delete "to state". Also is there a reference for MERRA underestimating temperatures in these regions? If not, please explain why it is plausible (i.e. SSWs).

P20 Figure 18 – middle panel colours are not explained (should they be blue and red?). Would also suggest using different colours for the bottom panel

Line 7 – Do you mean to say that MERRA temperature overestimates are a result WACCM overestimates of NO3? Instead of "consequently" do you mean "likewise"?

Line 10 – "mixing ratio values"

Line 12 – The sentence, "The very high. . . exponential function." Needs more discussion with quantification.

Figure 20 – I believe that left and right panels are switched (or incorrectly referenced in the caption). Also, I appreciate that all three panels have been plotted on the same scale, but this makes them more difficult to interpret. I recommend having the y-axes

on different scales. I would also highly recommend having the y-axes on a log scale. This would again make the figures and the discussion thereof clearer.

P21 First line – delete "ref."

P22 It is unclear here what the point of comparing the GOMOS and WACCM NO3/O3 ratio to theoretical calculations is. What does this tell us?

Line 5 – "comparison is done" should be "comparisons are done"

P23 Line 7 – "mesosphere below" should be "mesosphere just below"

Line 16 – what is meant by "to large extent"? When can and can't it be fitted to exponential function?

Lines 18-19 – I disagree that you've shown that you can use NO3 measurements as a proxy for SSWs. How did you show this? This would need more analysis and much more discussion. You would need to start by showing that deviations from the exponential curve only occur during SSWs.

Lines 21-22 – "physics and chemistry." should be "physics and chemistry is necessary."

---

## Author Comment (AC1) · 14 Mar 2018

We want to thank Reviewer 1 for his/her useful review of the paper. The answers to individual comments are shown below.

In addition to changes demanded by the reviewers we have updated all figures in order to increase their information content. In Figs. 5-7, 11-12, we have shown only 2 altitudes (earlier 3) for clarity. We have removed Fig. 14 because its content overlaps with Fig. 13. For Fig. 13 we have added also the WACCM-GOMOS difference plot. Figure 20 is redrawn. Instead showing the NO3/O3 ratio from theory, WACCM and GOMOS, we show the relative differences of this ratio from theory and WACCM to the ratio from
GOMOS. For readers' delight we have added one new figure (Fig. 21 of the paper, Fig. 1 in this response) that shows the vertical column differences between WACCM and GOMOS for our three gases.

We have changed our interpretation of the WACCM-GOMOS difference in the Arctic in the lower stratosphere. We assumed earlier that it could be a consequence of the NO2 increases from protons and downdrafts. Now the more plausible reason is that GOMOS sees larger ozone destruction during Arctic winters than what WACCM simulates. This can seen in new Figs. 6-7.

Specific answers to comments:

1. In equation (2), page 7, the scaling factor is computed using GOMOS data fkG(z). It may cause some problems when GOMOS values are small compared to their uncertainty with the problem of negative values within the error bars. Why not to use WACCM data fkW(z) instead?

Answer: We have experimented with both scaling factors. After a vote GOMOS was selected. In our analysis negative density values from individual occultations are included, but negative average values (from averaging over time and geolocation) are removed from the comparisons (both GOMOS and corresponding WACCM profiles). It does not make sense to include unphysical values for the comparison.

2. Page 7, line 13, from where is coming the factor 3x1.4826 for the elimination of outliers. Does it correspond to 3 sigma in the median statistic? This is not the same factor that the one given in equation (1) in Kyrölä et al. (2010a).

Answer: In the reference mentioned the quantity was the median-world 's analogue to the error of the mean. Now we are using median world's analogue to the 3*sigma-limit. For the sigma (standard deviation) a median absolute deviation, MAD, is used: MAD=median(|x-median(x)|). In order that MAD is a consistent estimator for the normal distribution, the MAD value needs to be multiplied by a factor 1.4826. For more detail
can be found from https://en.wikipedia.org/wiki/Median_absolute_deviation.

3. Page 7, lines 22-27, the discussion on regions defined with t_value > 2 is not clear. Please try to reformulate it.

Answer: This part is reorganised and rewritten.

4. Page 13, figure 8, do you have any explanation for the larger inter-annual NO2 variability in WACCM than in GOMOS? In general observations exhibit a larger variability than model outputs.

Answer: It is important to notice that these Sirius occultations take place only during August-September, most of the cases are from September. It is also important to note that the latitudinal region is partly inside the strong Antarctic vortex. In Fig.2 of this response we have shown three altitudes in this latitude region from all available stars (solid curves, GOMOS=blue, WACC=red) and the Sirius occultations by crosses (a gentle smoothing in time for all curves). Data cover from August 2002 to the end of September 2004. Around 2 hPa and above WACCM is usually slightly larger than GOMOS and this can be seen in Fig. 8 of the paper for 2002 and 2004. In August 2003 a remnant from an increased NO2-event during the summer 2003 is seen by a high peak in the two uppermost GOMOS curves and it temporarily lifts GOMOS above WACCM. When it returns back to the level below WACCM it crosses the WACCM level and this happens just at the time when Sirius measurements take place. Therefore WACCM and GOMOS agree during 2003 when Sirius data is used. We have modified the text after Fig. 8 as follows: The yearly variation in profiles and differences is large. Notice that the reason for this variation is the location of Sirius occultations near the Antarctic vortex where sporadic $NO_2$ enhancements are not totally contained in the polar latitudes.

5. Page 20, line 14, there is an extended discussion in Marchand et al. (2007) on the relation between GOMOS NO3 concentration and temperature with the same conclusion that NO3 is a good proxy for upper stratospheric temperature. Please cite this

paper: Reference: Marchand et al., 2007, Temperature retrieval from stratospheric O3 and NO3 GOMOS data, Geophys. Res. Lett., 34, L24809.

Answer: We apologise for this omission. Two articles by Marchand et al. (2004, 2007) have been added to references. We have also added a reference to Hakkarainen's thesis where the NO3-temperature relation was used in the assimilation.

6. Page 21, figure 20, it would be better to plot the NO3-temperature diagram with NO3 in log-scale in order to show the exponential relation.

Answer: A very good suggestion! The plots are now much more interesting. Because of the re-plotting, we discovered a bug in our software that caused corruption in the temperature data. All figures including temperature data are now corrected. There was also error in the caption. Hopefully everything is now correct in Fig. 20!
* * *
[Figure]

**Fig. 1.** The relative difference of WACCM and GOMOS vertical columns of ozone, NO2 and NO3.

[Figure]

0.907 hPa

1.87 hPa

21.8 hPa

Mixing ratio (ppb)

08 09 10 11 12 01 02 03 04 05 06 07 08 09 10 11 12 01 02 03 04 05 06 07 08 09 10

Month

**Fig. 2.** Response to comment 4 of the reviewer 1.

---

## Author Comment (AC2) · 14 Mar 2018

We want to thank Reviewer 2 for his/her elaborate review of the paper. The answers to individual comments are shown below.

In addition to changes demanded by the reviewers we have updated all figures in order to increase their information content. In Figs. 5-7, 11-12, we have shown only 2 altitudes (earlier 3) for clarity. We have removed Fig. 14 because its content overlaps with Fig. 13. For Fig. 13 we have added also the WACCM-GOMOS difference plot. Figure 20 is redrawn. Instead showing the NO3/O3 ratio from theory, WACCM and GOMOS, we show the relative differences of this ratio from theory and WACCM to the ratio from

[Figure]

GOMOS. For readers' delight we have added one new figure (Fig. 21) that shows the vertical column differences between WACCM and GOMOS for our three gases.

We have changed our interpretation of the WACCM-GOMOS difference in the Arctic in the lower stratosphere. We assumed earlier that it could be a consequence of the $NO_2$ increases from protons and downdrafts. Now the more plausible reason is that GOMOS sees larger ozone destruction during the Arctic winter than what WACCM simulates. This can seen in new Figs. 6-7.

Specific answers to comments:

Figures 1, 8, and 15 really should have a third panel that shows all the relative difference profiles (especially 8 and 15). Answer: These figures are now updated.

P1 Throughout the abstract, many value judgements are made (e.g. reasonable agreement, high correlation, etc.) without quantifying the values. Please backup these claims with the exact values that you have measured. Answer: Abstract changed.

Line 15 – unclear what is meant by "the validity region" Answer: Validity region in altitude. Text changed.

P2 Lines 5-6 – please briefly explain how these references are examples of improving understanding of accuracy. Answer: text changed.

Line 14 – Smith et al. 2013 (doi:10.1002/jgrd.50445) would be an excellent ref to include as well. Answer: Added.

Lines 14-16 – Mention of validation and comparison studies should include references. Answer: Reference to Hubert et al. (2016) added.

Line 22 – "see" on its own tends to imply a full list, something like "e.g." or "see for example" would be more appropriate Answer: Text changed.

Line 24 – should mention this is at all altitudes. Answer: Text changed.

Line 26 – change to "to the lower thermosphere" Answer: Text changed.

Line 32 – Does this mean that ion chemistry is included? Or just (non ion) chemistry within the D region? Please specify in the text. Answer: Yes, text changed.

P3 Lines 1-5 – tends to be vague. Terms "reasonable agreement," "compares well," and "found to be similar" need specific quantified values in order to back up these judgements. Same with "good" on line 11. Answer: We have now added WACCM total ozone bias numbers from Eyring, V., Shepherd, T. G., and W., W. D., eds.: SPARC CCMVal Report on the Evaluation of Chemistry-Climate Models, vol. No. 5. We removed the statement about WACCM and HALOE, because we could not find any numbers or even deviation plots from literature, only messy plots with thousand or more models included!

Line 11 – What is meant by brightest? Answer: Measured by the apparent magnitude at Earth's position.

Line 16 – "mesosphere and" can be deleted Answer: Text changed.

Line 17 – "comparison" should be "comparisons" Answer: Text changed.

Line 27 – "Sec. the" should be "Sec. 3 the" Answer: Text changed.

P4 Lines 1-2 – should be "this approach has" Answer: Text changed.

Line 6 – "in detail" is not needed Answer: Text changed.

Line 7 – "those" is not necessary Answer: Text changed.

Line 12 – should be something more like "there is an ozone-specific flag that screens out stars. . ." Answer: Text changed.

Line 14 – should be "outliers" and "stratosphere" Answer: Text hanged.

Lines 14-15 – I assume you're not setting the flags to zero, rather you're only using profiles where flag values are zero. Answer: Text changed.

Line 27 – do you mean "within ±3%" ? Answer: Text changed.

Line 30 – do you mean "within ±4%" ? Answer: Text changed.

P5 5th line (labelled l38) – could reiterate that this is nighttime profiles being compared. Answer: Text changed.

Line 9 – Sheese et al. 2016 (doi:10.5194/amt-9-5781-2016) is the more recent ACE-FTS NOy validation reference and should be added. Seems to show GOMOS âĹij0-10% higher between 23-30 km, âĹij25% higher at 30-45 km (although seems ACE-FTS has low bias of âĹij10% in this region). Answer: Text updated.

Line 25 – what kind of observations? Answer: Historical surface concentrations of greenhouse gases were taken from Meinshausen et al. (2011)". Meinshausen, M., and Coauthors, 2011: The RCP greenhouse gas concentrations and their extensions from 1765 to 2300. Climatic Change, 109, 213–241, doi:10.1007/s10584-011-0156-z. This reference added and text changed.

P6 Line 16 – delete "a" Answer: Text changed.

Line 19 – would be good to add here how many collocated profiles there are Answer: Text changed

P7 Eqn 2 – could change to "100%" Answer: Changed

Line 6 – "WACCM" should be "GOMOS" Answer: Yes, text changed

Line 10 – ". . .processes while keeping reasonable. . ." Line 11 – is it the Pearson correlation coefficient?ÂĺAnswer: Yes, it is Pearson. Text changed.

Line 12 – What is meant by "averages over number of"? Answer: Text hanged.

Lines 13-14 – The equation doesn't make it clear exactly how the data is being filtered. A please rephrase for clarity. Is this method done using all data at a given altitude? Is it done in latitude bins? Also, in atmospheric datasets where the data is very often

neither Gaussian nor uni-modal, using the MAD as a filter can often lead to filtering out a lot of inlying data with the outlying data. If you haven't already, please check that this method isn't "over-filtering" your data, and if everything is okay it would be good to specify that this check was done and how much data is filtered out using this method. Answer: This part is reorganised and partly rewritten. For ozone the outlier filter removes on average 1 % of measurements. At the ozone minimum altitude and in the polar regions the filtering effect is larger, up to 5 %. For NO2 and NO3 the filtering is from 1% (non-polar) to 5% (polar regions). We consider these numbers acceptable. Anyhow, in this paper our focus is on differences of paired data between GOMOS and WACCM, not so much 100% mapping of the atmosphere with all its diversity. The sparse sampling of GOMOS measurements does not support this kind of dream.

P8 Figure 1 – It would be nice if both figures had the same y-axis labels Answer: Changed. We removed the km-axis as we added the third sub-panel requested by the reviewer. Available space disappeared.

Caption – Please specify in caption that these are Aug-Sept profiles Answer: Text hanged.

Discussion of Figure 1 – It would be worth noting that both GOMOS and WACCM are exhibiting the tertiary peak and are in good agreement in both height (âĹij68 km, which actually seems a bit low for the tertiary peak, see Degenstein et al 2005, doi:10.1016/j.jastp.2005.06.019) and concentration. Answer: Some discussion of the tertiary peaks is done after Fig.3. and the reference added. In this kind of mission average the tertiary peak is not so clearly visible and in our mind it does not deserve detailed discussion.

Line 13 – delete "the" Answer: Changed.

P9 Line 1 – "reaches again 2%" must be a typo. At the secondary peak, the differences clearly much larger. Answer: No, it is correct. If w use the words from the mean-world we are saying here that the error of the mean averaged over the 10 year mission is 2%.

The standard deviation during each year is of course much larger but the mean is quite precise.

Fig 2 caption – might want to say "A cell with a dot marks where there are no collocated profiles." Same with Figures 3, 4, 9, 10, 16, and 17. Answer: Text changed. But should it be like "A cell with a dot marks a point where there are no collocated profiles"?

P10 Line 18 – "in the two cases" Answer. Text changed.

Line 30 – it may be more prudent to something more along the lines of "we have not been able to identify any potential sources of uncertainty that could lead to such a large error in the GOMOS data" Answer: Text changed.

P11 Figure 5 – At the secondary maximum the WACCM seasonal variation is very difficult to discern and it's not immediately clear that WACCM and GOMOS are in phase. Could this panel have the y-axis on a log scale? Answer: Changed. And we have reduced the displayed altitudes from 3 to 2 in order to improve the clarity of the figure.

P12 Figures 6 – A legend (maybe on the rhs) would make the plots much easier to read. Same with Figures 7, 11, 12, 18. Answer: Legends added.

Figure 7 – The red and blue is slightly confusing, because the reader will naturally be comparing to Figure 6 where the same colours indicate only Arctic values. Please use two styles of lines that relate better to Figure 6. Same with Figure 12. Answer: We have added legends, it probably helps. In many figures we are suffering from the availability of clearly differing colours. The only help for a reader is to enlarge plots on the computer screen!

Line 34 – "correlation is typically very high," Answer: The changed.

Lines 34-35 – I find this sentence somewhat confusing, please rephrase for better clarity. Answer: Text changed slightly.

P13 First paragraph – please include quantification of the differences and correlation

for all three panels of Figure 5. Answer: Text changed.

Line 4 (6th line) – "whereas" Answer: Text changed.

Lines 5-12 – Please include quantification of the differences and correlation for all three panels of Figure 6 and 7. Answer: Text changed.

Figure 8 – should include altitude scale to match Fig 1. Same with Figure 15. Answer: Because we added third panel, we had to remove all the km-scales.

P14 First paragraph – are these also Aug-Sept? If so, please mention here and in Fig 8 caption. Same with NO3 results/figure. Answer: Text changed.

Fourth line – "maximum at 5 hPa the difference is within" Answer: Text changed.

Line 9 – "will be discussed" Answer: Text changed.

Line 11 – "is typically 0-10%" and "typically agree within $\pm$5%" as differences do reach higher values in the respective regions Answer: Text changed.

P15 First paragraph – Please quantify discussion of correlation coefficients Answer: Text changed.

Line 13 – Should start sentence with something like, "As seen in Fig. 13 and 14,. . ." Answer: Text changed.

P16 First line – it's somewhat confusing that you're discussing the difference in ppb when figure 13 is in %, please make this consistent (or discuss both % and ppb). Unless, you're referring to Fig 14 here, but Fig 14 doesn't show results for 0.5 hPa. Either way, this section needs to be made clearer (as to what Figures you're discussing and what they show). Answer: In Fig. 13 we now show both GOMOS and WACCM, and mixing ratio absolute difference. Fig. 14 is now removed. Text updated.

Line 5 – "peak density, âĹij2hPa" Answer: Text changed.

Line 10 – "is typically inside" Answer: Text changed.

P17 Figure 13 – units are missing on both panels Answer: Added.

Last line – "The secret behind" could be something more like "the reason for" Answer: Text changed.

P18 Fifth line – delete "to state". Also is there a reference for MERRA underestimating temperatures in these regions? If not, please explain why it is plausible (i.e. SSWs). Answer: We have mellowed the text.

P20 Figure 18 – middle panel colours are not explained (should they be blue and red?). Would also suggest using different colours for the bottom panel Answer: Changed. We have added line legends.

Line 7 – Do you mean to say that MERRA temperature overestimates are a result WACCM overestimates of NO3? Instead of "consequently" do you mean "likewise"? Answer: We mean that MERRA temperature overestimates are the probable source of WACCM NO3 overestimates. Text improved.

Line 10 – "mixing ratio values" Answer: Text changed.

Line 12 – The sentence, "The very high. . . exponential function." Needs more discussion with quantification. Answer: We removed this sentence because there are not enough cases where this statement is true.

Figure 20 – I believe that left and right panels are switched (or incorrectly referenced in the caption). Also, I appreciate that all three panels have been plotted on the same scale, but this makes them more difficult to interpret. I recommend having the y-axes on different scales. I would also highly recommend having the y-axes on a log scale. This would again make the figures and the discussion thereof clearer. Answer: Yes, yes, the caption is wrong! Thanks for your keen eye. We have now plotted using log-scale. The plots are now much more interesting. Because of the re-plotting, we discovered a bug in our software that caused corruption in the temperature data. All figures including temperature data are now corrected. Hopefully everything is now correct in Fig. 20!

P21 First line – delete "ref." Answer: Text changed.

P22 It is unclear here what the point of comparing the GOMOS and WACCM NO3/O3 ratio to theoretical calculations is. What does this tell us? Answer: Our approach is this: A complicated model like WACCM is entirely a numerical machine with massive amounts of approximations and parametrisations. If there are results from chemical and physical theories that numerical models should fulfil, we need to check if they do. Because these kind of check point results are not necessarily universally true, but assume some additional conditions (in our case chemical equilibrium), it is important also to see if they are obeyed by experimental measurements.

Line 5 – "comparison is done" should be "comparisons are done" Answer: Text changed.

P23 Line 7 – "mesosphere below" should be "mesosphere just below" Answer: Text changed.

Line 16 – what is meant by "to large extent"? When can and can't it be fitted to exponential function? Answer: Changed to "fitted reasonably well".

Lines 18-19 – I disagree that you've shown that you can use NO3 measurements as a proxy for SSWs. How did you show this? This would need more analysis and much more discussion. You would need to start by showing that deviations from the exponential curve only occur during SSWs. Answer: You are correct. Because this topic is not in the focal point of the paper we leave further analysis to future publications. We have downshifted our text.

Lines 21-22 – "physics and chemistry." should be "physics and chemistry is necessary." Answer: Text changed.
* * *
[Figure]

**Fig. 1.** The The relative difference of WACCM and GOMOS vertical columns of ozone, NO2 and \NO3.

---

## Author Response (AR1)

Title: Middle atmospheric ozone, nitrogen dioxide, and nitrogen trioxide in 2002--2011: SD-WACCM simulations compared to GOMOS observations
Author(s): Erkki Kyrölä et al.
MS No.: acp-2017-1161
MS Type: Research article
Iteration: Revised Submission
Special Issue: Quadrennial Ozone Symposium 2016 – Status and trends of atmospheric ozone (ACP/AMT inter-journal SI)

**Author's response**

**1. Figures:**
We have submitted our replies to the reviewers on 14 March 2018 where we have addressed all points raised by the reviewers.  The remarks were mainly concerning the style of our expression. Both reviewers suggested that we should use log-scale in Fig. 20 (now Fig. 19). This change revealed a small bug in our processing software and we had to correct also Figs. 19 and 21.

We have updated all figures in order to increase their information content. The following larger changes were done (figure numbers refer to the original submission):
-Figs. 1, 8, 15: Relative difference panel added.
-Figs. 2-4, 9-10, 16-17: Contour lines  were added. For correlation plots
        significance data were added.
-Figs. 5-7, 11-12, we have shown only 2 altitudes (earlier 3) for clarity.
-Fig. 13 we have added also the WACCM-GOMOS difference plot.
-Fig. 14 is removed because its content overlaps with Fig. 13.
-Fig. 19 is redrawn using log-scale
-Fig. 20 is redrawn. Instead showing the NO3/O3 ratio from theory,  WACCM
        and GOMOS, we show now the relative differences of this ratio
        from theory and WACCM with respect to the ratio from GOMOS.

We have added one new figure (now Fig. 21 of the new paper) that shows the vertical column differences between WACCM and GOMOS for our three gases. This figure summarizes difference results in a concise way.

2. **Analysis**
We have changed our interpretation of the WACCM-GOMOS difference in the Arctic in the lower stratosphere. We assumed earlier that it could be a consequence of the NO2 increases from solar storms and downdrafts. Now the more plausible reason is that GOMOS sees larger ozone destruction during Arctic winters than what WACCM simulates.

We have added in Sec. 6 a summary of the Antarctic NO2 increases during June-September 2003.

**3. Methods**

We have made a few changes in our data processing in order to make it more straightforward. These are:

-All results are now based on 5-day time series. Monthly time series are removed.

- The minimum number of measurements needed for time series is now 10 for each star. Earlier we had separate limits for each star and combination of stars for averages.

**4. Text**

The differences in text are numerous and visible in the included difference file. Changes include the ones proposed by the reviewers (some of them are no more relevant), but most of the changes are from authors. The main reason for textual changes is the need to increase quantitative information about our results.  We had over 900 000 measured and simulated trace profiles under inspection!

**5. Formal**

There is change in the address of the first author (and his colleagues) because of the change of the Finnish Meteorological Institute organisation.

[revised manuscript text omitted]